METABETA

# A FAST NEURAL MODEL FOR BAYESIAN MIXED-EFFECTS REGRESSION

## ABSTRACT

Hierarchical data with multiple observations per group is ubiquitous in empirical sciences and is often analyzed using mixed-effects regression. In such models, Bayesian inference gives an estimate of uncertainty but is analytically intractable and requires costly approximation using Markov Chain Monte Carlo (MCMC) methods. Neural posterior estimation shifts the bulk of computation from inference time to pre-training time, amortizing over simulated datasets with known ground truth targets. We propose `metabeta`, a transformer-based neural network model for Bayesian mixed-effects regression. Using simulated and real data, we show that it reaches stable and comparable performance to MCMC-based parameter estimation at a fraction of the usually required time.

## 1 INTRODUCTION

Much of the data we work with has a hierarchical structure that naturally clusters into subgroups. When predicting the efficacy of a drug, for example, there may be subpopulation-specific effects in addition to effects on the population as a whole. When building a movie recommendation system, some films may be universally popular, yet individual preferences still matter. When looking at plant growth, the same fertilizer may perform well in one field but poorly in another due to local conditions. These challenges can be addressed using mixed-effects models, which provide a principled framework for capturing both overall trends (fixed effects) and group-specific deviations (random effects). Mixed-effects models have been widely adopted across disciplines – including ecology, psychology, and education – and are by now considered a standard approach for analyzing hierarchical data (Gelman & Hill, 2007; Harrison et al., 2018; Gordon, 2019; Yu et al., 2022).

In many such applications, we would like to estimate the parameters of a mixed-effects model in a Bayesian manner, enabling the incorporation of prior knowledge and the explicit quantification of uncertainty (Figueroa-Zúñiga et al., 2013; Gelman et al., 2013). However, closed-form solutions are generally unavailable even for the simplest cases, necessitating computationally expensive approximate inference methods such as Markov Chain Monte Carlo (MCMC, Metropolis et al., 1953). From a practitioner's perspective, this is undesirable, as MCMC typically requires significant runtime, even for moderately sized datasets.

In this work, we introduce `metabeta`, a probabilistic neural network model, that is designed to efficiently approximate Bayesian inference for mixed-effects regression. It is trained via neural posterior estimation (Rezende & Mohamed, 2015; Gordon et al., 2018; Wildberger et al., 2023; Hollmann et al., 2025), amortizing computation costs over many simulated hierarchical datasets with available ground truth parameters. We demonstrate that `metabeta` achieves accuracy comparable to Hamiltonian Monte Carlo (HMC), which is the gold-standard MCMC method for Bayesian mixed-effects regression (Neal, 2011; Bürkner, 2018; Capretto et al., 2022). Importantly, our model reduces inference time by orders of magnitude, thereby greatly broadening the range of feasible applications for Bayesian mixed-effects regression. To further facilitate `metabeta`'s adoption for rapid deployment and plug-and-play compatibility, we provide open-source Python code for our implementation and plan to release a package with pretrained models that integrates seamlessly with existing analysis pipelines (Bürkner, 2018; Abril-Pla et al., 2023).

## 1.1 RELATED WORK

Neural posterior estimation (NPE) — the simulation-based amortization of a neural network posterior — has a long and well-established history. Early work by Papamakarios & Murray (2016) introduced neural conditional density estimators for directly approximating posteriors from simulations. This approach was extended by Lueckmann et al. (2017), who incorporated importance weighting to enable sequential refinement of posterior approximations, and by Greenberg et al. (2019), who proposed automatic posterior transformation, increasing flexibility in proposal adaptation and posterior modeling. These methods form the core foundations of amortized simulation-based Bayesian inference.

Rezende & Mohamed (2015) pioneered the use of conditional normalizing flows (Papamakarios et al., 2021; Kobyzev et al., 2021) for amortized inference. Together with Gordon et al. (2018), they laid the groundwork for BayesFlow (Radev et al., 2020; 2023), which introduced a practical workflow for globally amortized Bayesian inference using summary encoders and normalizing flows. Subsequent extensions adapted BayesFlow to hierarchical Bayesian models (Habermann et al., 2024) and to non-linear mixed-effects models for cell biology and pharmacology (Arruda et al., 2023). In both cases, the prior distribution is fixed at training time, requiring retraining whenever a user wishes to change the prior. This off-loads the amortization process to potential end-users, which strongly diminishes the runtime advantage of NPE for practical purposes.

More recently, transformer-based architectures have emerged as a distinct line of research for amortized Bayesian inference. For instance Distribution Transformers (Whittle et al., 2025) represent prior and posterior as Gaussian Mixture Models whose parameters are mapped by transformers. A thorough comparison of transformer-based NPE methods was recently conducted by Mittal et al. (2025). These works demonstrate that transformer-based NPE can adapt efficiently to varying priors and heterogeneous datasets. However, they have not been tailored specifically to mixed-effects regression, and explicit incorporation of priors in NPE remains an active field of research.

## 2 METHODS

We briefly formalize mixed-effects regression (Section 2.1) and define a synthetic distribution over hierarchical datasets representative of scenarios practitioners care about (Section 2.2). We then present a neural network architecture that takes an entire dataset and priors as inputs and returns posterior distributions over all regression parameters (Section 2.3). This model is trained on synthetic datasets with available ground truth to perform accurate posterior inference (Section 2.4). In a final post-training step, we refine the model's outputs using importance sampling and conformal prediction (Section 2.5). All our code is implemented in `PyTorch` 2.7.1 (Paszke et al., 2019) and openly available at `https://github.com/censored-for-review`.

## 2.1 MIXED-EFFECTS REGRESSION

Mixed-effects regression extends traditional regression by explicitly accounting for within-group dependency in hierarchical data (Gelman & Hill, 2007; Brown, 2021; Fahrmeir et al., 2013). To model this dependency, mixed-effects regression distinguishes between two effect types:

- **Fixed effects** $\boldsymbol{\beta} \in \mathbb{R}^d$ capture the general, group-independent relation between predictor variables $\mathbf{X}_i \in \mathbb{R}^{n_i \times d}$ and the regression output variable $\mathbf{y}_i \in \mathbb{R}^{n_i}$.
- **Random effects** $\boldsymbol{\alpha}_i \in \mathbb{R}^q$ capture additional, group-specific variations for $q \leq d$ predictors. For each group $i = 1, \ldots, m$, we treat $\boldsymbol{\alpha}_i$ as samples from $\mathcal{N}_q(\mathbf{0}, \mathbf{S})$[1]

This yields the model:

$$\mathbf{y}_i = \mathbf{X}_i \boldsymbol{\beta} + \mathbf{Z}_i \boldsymbol{\alpha}_i + \boldsymbol{\varepsilon}_i \,, \tag{1}$$

with independent additive noise $\boldsymbol{\varepsilon}_i \sim \mathcal{N}_{n_i}(\mathbf{0}, \sigma_\varepsilon^2 \mathbf{I}_{n_i})$. The random effect predictor matrix $\mathbf{Z}_i$ is typically a submatrix of $\mathbf{X}_i$.

---

[1]$\mathbf{S} \in \mathbb{R}^{q \times q}$ is generally symmetric positive-definite, but for practical purposes it is often additionally constrained to be diagonal and we include this constraint in our model.

Note, that the $n_i$ observations are *conditionally independent* given some fixed $\boldsymbol{\alpha}_i$ but *marginally dependent* over $\boldsymbol{\alpha}_i$:

$$\mathbf{y}_i|\boldsymbol{\alpha}_i \sim \mathcal{N}_{n_i}(\mathbf{X}_i\boldsymbol{\beta} + \mathbf{Z}_i\boldsymbol{\alpha}_i,\ \sigma_\varepsilon^2\mathbf{I}_{n_i}) \quad \Longrightarrow \quad \mathbf{y}_i \sim \mathcal{N}_{n_i}(\mathbf{X}_i\boldsymbol{\beta},\ \mathbf{Z}_i\mathbf{S}\mathbf{Z}_i^\top + \sigma_\varepsilon^2\mathbf{I}_{n_i}).$$

The goal of Bayesian mixed-effects modeling is to obtain posteriors for all unobserved global $(\boldsymbol{\beta}, \sigma_\varepsilon^2, \mathbf{S})$ and local $(\boldsymbol{\alpha}_i)$ regression parameters, conditioned on the observed predictors, outcomes and priors of the global parameters.

## 2.2 Data Simulation and Preprocessing

To train our neural posterior estimator, we simulate hierarchically structured datasets as shown in Figure 1A.

*Priors*: For each dataset, we sample hyper-parameters that specify each multidimensional prior. That is, for $d$ fixed effects, we first sample a $d$-dimensional prior, from which the $d$ fixed effects are sampled later.

*Regression parameters*: We use the default prior families of `Bambi` (Capretto et al., 2022). (1) $q$ random effect variance parameters are sampled from half-normal distributions, (2) then, $m \times q$ random effect vectors are sampled from a diagonal Gaussian using these variance parameters. Independently, $d$ fixed effects are sampled from another diagonal Gaussian. (3) Noise variance is sampled from a half-$t$-distribution, and then independent noise is sampled from a normal distribution with this variance.

*Observations*: The predictors $\mathbf{x}_{ij}$ are sampled from two sources: Synthetic distributions and real datasets. The random effects predictors are set to $\mathbf{z}_{ij} = \mathbf{x}_{ij}$ for $j \leq q$ and 0 otherwise. Predictors are standardized and passed through equation 1 with the regression parameters and noise to generate outcomes $\mathbf{y_i}$ for each group $i$. Further details on the simulation procedure can be found in Appendix A.

For the test sets, we use `Bambi` on top of `PyMC` (Abril-Pla et al., 2023) to estimate all posteriors with the No-U-Turn sampler (a variant of HMC) (Hoffman & Gelman, 2011). We run four chains with 2500 tuning iterations and 1000 posterior draws each. For the MCMC model, we supply the true priors and the generative model used in the simulation. For a fair comparison, we exclude datasets with divergent MCMC samples from the test set. We additionally fit a variational inference (VI) approximation of the probabilistic model (Kucukelbir et al., 2016; Blei et al., 2017), which is typically the preferred computationally cheaper alternative to MCMC. We use the same model specification and the ADAM optimizer Kingma & Ba (2017) with learning rate $\eta = 0.005$, 50000 training iterations and 4000 draws. MCMC and VI fit diagnostics are included in Appendix F.

## 2.3 Model Architecture

The model architecture takes inspiration from `BayesFlow` (Radev et al., 2020; Habermann et al., 2024) and `TabPFN` (Hollmann et al., 2025) and has two main parts: (1) a *summary network* that computes a maximally informative dataset statistic over observations, and (2) a *posterior network* that uses the summary and priors to propose a joint posterior over regression parameters. Both are trained end-to-end. Since mixed-effect datasets are hierarchically structured, we use two summary and posterior networks, one for the global parameters (fixed effects and variance parameters) and one for the local parameters (group-specific random effects). The training and inference pipeline is visualized in Figure 1B. Data preprocessing is detailed in Appendix B and Appendix C.

### Summary Network

Datasets vary in the number of groups and observations per group. A summary network $f_\Sigma$ extracts information for the posterior by pooling over all instances in a dataset. Since the data is structured hierarchically, it needs to be summarized accordingly over all exchangeable instances: In a first step, we pool over observations per group, generating $m$ local summaries $\mathbf{s}_1, \ldots, \mathbf{s}_m$. In a second step, we pool the local summaries over groups, generating a global summary $\mathbf{s}$. For summarization, we opted for a set transformer (Lee et al., 2019). Our implementation consists of multiple transformer encoder blocks (Vaswani et al., 2017), followed by averaging over the resulting sequence of transformer

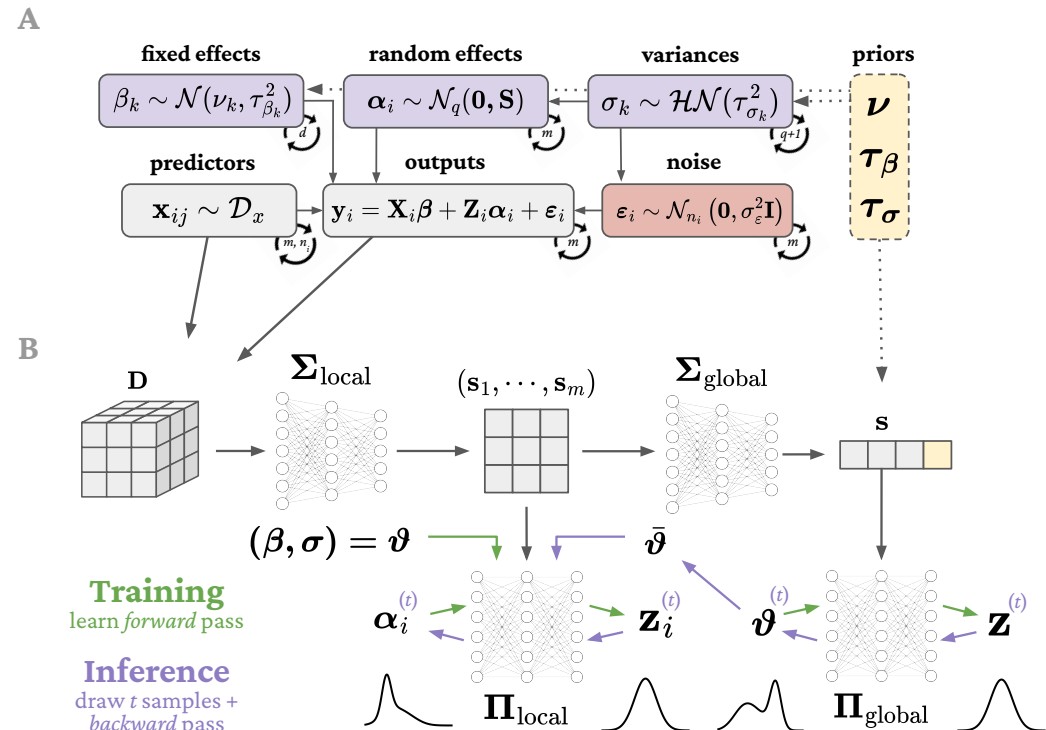

**Figure 1:** (A) *Dataset Simulation*. Given a set of priors, we sample regression parameters and noise in a cascading way. Predictors are sampled from various distributions for training and from real datasets for testing, and outcomes are generated according to equation 1. (B) *Model Pipeline*. Observed data are summarized locally (per group) and globally (across groups). During training, the posterior networks learn the forward mapping from the true regression parameters to a simple multivariate base distribution, conditioned on the respective summaries and priors. During inference, we draw *k* samples from the base distribution, and apply the implicitly learned backward mapping to them, approximating sampling from the unknown target posterior.

outputs. This yields the important property of permutation invariance, i.e. the summary stays the same regardless of the input ordering along the sequence dimension. The local and global summary network both consist of 3 transformer encoder blocks with 128 units, equally large feedforward layers, 8 attention heads, 1% dropout and GELU activations (Hendrycks & Gimpel, 2023).

POSTERIOR NETWORK

Posterior networks $f_\Pi$ take the dataset summaries and priors as inputs and propose a joint posterior for a set of parameters. Inference on global and local parameters is separated for hierarchical NPE (Rodrigues et al., 2021; Heinrich et al., 2023; Habermann et al., 2024). Inference on global parameters $\vartheta = (\beta, \mathbf{S}, \sigma_\varepsilon^2)$ is conditioned on the global summary and the parameter priors. Inference on local variables $(\alpha_i)$ is conditioned on the separate local summaries and the global parameters (the true ones during training, and the inferred ones during validation). We opted for a normalizing flow (Papamakarios et al., 2021) as our posterior network:

A normalizing flow learns an invertible mapping from a $d$-dimensional random variable $\mathbf{z}_n$ with a complex distribution to a $d$-dimensional random variable $\mathbf{z_0}$ with a regular distribution (e.g. a multivariate normal). The flow consists of a finite composition $T$ of continuously differentiable and invertible transforms $T_i$ with triangular Jacobians, $T = T_n \circ \cdots \circ T_1$. For some random variable $\mathbf{z}_0 \sim \mathcal{N}_d(\mathbf{0}, \mathbf{I})$, we model $T(\mathbf{z}_n) = \mathbf{z}_0 \iff T^{-1}(\mathbf{z}_0) = \mathbf{z}_n$ with $p_n(\mathbf{z}_n) = p_0(\mathbf{z}_0) |\det J_T(\mathbf{z}_0)|$. Each invertible transform $T_i$ is parameterized by a neural network that takes part of the current hidden state $\mathbf{z}_t$ and the summary $\mathbf{s}$ as inputs. Because of their efficiency, we opted for conditional affine coupling as our normalizing flow architecture (Dinh et al., 2014; 2017). For the base distribution we use a diagonal multivariate location-scale $t$ distribution with learnable parameters for each dimension

(Alexanderson & Henter, 2020). For both posterior networks, we use 8 affine coupling flow blocks parameterized by MLPs with three 256-unit feedforward layers, skip connections (He et al., 2016), 1% dropout and ReLU activations.

## 2.4 LEARNING

To calculate the loss for the global parameters, we use the forward Kullback-Leibler divergence between the unknown true posterior $p(\boldsymbol{\vartheta}|\mathbf{s})$ and its flow-based approximation $p_\Pi(\boldsymbol{\vartheta}|\mathbf{s}) := p_n(\mathbf{z}_n|\mathbf{s})$,

$$\ell_\Pi(\boldsymbol{\vartheta}, \mathbf{s}) \propto -\mathbb{E}_{\boldsymbol{\vartheta}, \mathbf{s}}\left[\log p_\Pi(\boldsymbol{\vartheta}|\mathbf{s})\right] = -\mathbb{E}_{\boldsymbol{\vartheta}, \mathbf{s}}\left[\log p_0(T(\boldsymbol{\vartheta}|\mathbf{s})) + \log|\det J_T(\boldsymbol{\vartheta}|\mathbf{s})|\right],$$

where $T$ and thereby the approximation $p_\Pi(\boldsymbol{\vartheta}|\mathbf{s})$ depend on the posterior network. Since the summary $\mathbf{s}$ of data $\mathbf{D}$ is itself depending on the summary network, the end-to-end loss can be written as

$$\ell_{\Pi, \Sigma}(\boldsymbol{\vartheta}, \mathbf{D}) \propto -\mathbb{E}_{\boldsymbol{\vartheta}, \mathbf{D}}\left[\log p_\Pi(\boldsymbol{\vartheta}|f_\Sigma(\mathbf{D}))\right] = -\mathbb{E}_{\boldsymbol{\vartheta}, \mathbf{D}}\left[\log p_0(T(\boldsymbol{\vartheta}|f_\Sigma(\mathbf{D})) + \log|\det J_T(\boldsymbol{\vartheta}|f_\Sigma(\mathbf{D}))|\right].$$

The objective for the local parameters is completely analogous. We sum the local losses over groups and add the result to the global loss, which follows a potential factorization of the joint posterior over both types of regression parameters (see Appendix D). The expectation is approximated by averaging over the batch. Model weights are updated using the Schedule-Free AdamW optimizer (Defazio et al., 2024). We train separate models for different numbers of fixed effects and random effects until convergence, which requires between $10^5$ and $10^6$ training sets in our case.

## 2.5 POST-HOC REFINEMENT

### IMPORTANCE SAMPLING

In the idealized limit of infinite network capacity, neural posterior flexibility, infinite simulations, and perfectly converged optimization, our model would not require any further correction. However, in practice these conditions are never fully met. Learning $p_\Pi$ by minimizing the forward Kullback-Leibler Divergence naturally forces $p_\Pi$ to be positive wherever $p$ is positive, making $p_\Pi$ mass-covering (Jerfel et al., 2021). Thus, we can use importance sampling to improve posterior estimation (Tokdar & Kass, 2010; Dax et al., 2023) to correct for inaccuracies of the amortized estimator. For each sample $\boldsymbol{\vartheta}_k \sim p_\Pi(\boldsymbol{\vartheta}|\mathbf{D})$ we assign an importance weight,

$$w_k = \frac{p(\mathbf{D}|\boldsymbol{\vartheta}_k)p(\boldsymbol{\vartheta}_k)}{p_\Pi(\boldsymbol{\vartheta}_k|\mathbf{D})},$$

which is well-defined, as $p_\Pi$ is only zero if the numerator is zero. We use the weights to refine statistics of the samples (e.g. the posterior mean or empirical CDFs). Since we have two posterior networks and the local posterior is conditioned on the global estimates, we perform alternating importance sampling for both. For more details, please see Appendix E.

### CALIBRATION WITH CONFORMAL PREDICTION

Uncertainty quantification is a hallmark of Bayesian inference, making the fidelity of the approximate posterior's credible intervals a critical concern. Posterior samples can be used to calculate empirical quantiles and thus also intervals that contain $c\%$ of the posterior density. Due to the mass-covering property of $p_\Pi$, the learned posteriors tend to be too wide – i.e. the true parameter is inside the $c\%$ credible interval in more than $c\%$ of the cases. This can be quantified with the coverage error

$$\mathrm{CE}(\alpha) = \frac{1}{B}\sum_{b=1}^{B}\mathbb{1}\left(\vartheta^{(b)} \in C_\alpha^{(b)}\right) - (1 - \alpha),$$

which should asymptotically approach 0 under perfect coverage. Too liberal coverage is a commonly known issue of normalizing flows (Chen et al., 2025; Dheur & Taieb, 2025). Conformal prediction (Vovk et al., 2022; Shafer & Vovk, 2008; Angelopoulos & Bates, 2022) is a general-purpose method that constructs distribution-free prediction sets $\hat{C}_\alpha$ such that $\mathbb{P}(\boldsymbol{\vartheta} \in \hat{C}_\alpha) \geq 1 - \alpha$. To construct $\hat{C}_\alpha$, we use a calibration set to calculate the difference between the true $\boldsymbol{\vartheta}$ and the closest border of the proposed $1 - \alpha$ credible interval $C_\alpha$. The $1 - \alpha$ quantile of these differences is then added to the proposed interval borders, widening them if the value is positive and narrowing them otherwise. Importantly, this does not require retraining but efficiently refines credible intervals post-hoc.

## 3 RESULTS

We test our model against HMC on a toy dataset with highly constrained parameters and uncorrelated normal data (Section 3.1), in-distribution test sets with predictors $\mathbf{X}$ sampled from real datasets (Section 3.2), and out-of-distribution test sets containing subsets of real data where the parameters $\vartheta$ are unknown and the outcomes $\mathbf{y}$ are kept original (Section 3.3). Each test set has a batch size of $128$ regression datasets with varying numbers of observations and signal-to-noise ratios.

We use the following evaluation metrics: We quantify the *posterior predictive accuracy* with the negative log likelihood (NLL), $-\log p(\mathbf{y}|\hat{\vartheta})$, which measures how well the fitted model describes the observed data. We calculate the mean NLL over sampled parameters and the median over the test batch. We gauge *parameter recovery* (true parameters vs. posterior means) with Pearson's correlation $r$ and RMSE. We check *posterior calibration* by averaging coverage errors over a set of alpha levels, $\mathrm{CE} = \frac{1}{|A|}\sum_{\alpha \in A}\mathrm{CE}(\alpha)$. Median run times per single dataset are measured in seconds on a MacBook Air M2 with 24GB of RAM using Metal Performance Shaders (MPS) where possible.

We gauge simulation-based calibration (SBC, Talts et al., 2020; Säilynoja et al., 2022; Deistler et al., 2025) using empirical CDF plots, comparing the parameter rank statistics against a theoretical uniform CDF. Finally, we plot posterior predictive distributions (Gelman et al., 2013), which visualize how much the posterior predictive samples $\tilde{\mathbf{y}}_{\mathbf{t}} \sim p(\mathbf{y}|\hat{\vartheta}_{\mathbf{t}})$ match the actual $\mathbf{y}$. All metrics are calculated for `metabeta`, HMC and VI posterior samples.

### 3.1 TOY EXAMPLE

To gauge if the pipeline works for both our model and HMC, we first test both on a toy example with $d = 2$ and $q = 1$, where the observed single predictor is sampled from a standard normal distribution. Result figures can be found in Appendix G. All models reach almost perfect parameter recovery correlation for fixed effects ($r = 0.999$ each), variance parameters ($r_{\mathrm{MB}} = 0.995$ vs. $r_{\mathrm{HMC}} = 0.991$ vs. $r_{\mathrm{VI}} = 0.991$), and random effects ($r = 0.959$ each). The same pattern arises for recovery error for fixed effects ($\mathrm{RMSE}_{\mathrm{MB}} = 0.023$ vs. $\mathrm{RMSE}_{\mathrm{HMC}} = 0.021$ vs. $\mathrm{RMSE}_{\mathrm{VI}} = 0.030$), variance parameters ($\mathrm{RMSE}_{\mathrm{MB}} = 0.022$ vs. $\mathrm{RMSE}_{\mathrm{HMC}} = 0.025$ vs. $\mathrm{RMSE}_{\mathrm{VI}} = 0.034$), and random effects ($\mathrm{RMSE}_{\mathrm{MB}} = 0.108$ vs. $\mathrm{RMSE}_{\mathrm{HMC}} = 0.108$ vs. $\mathrm{RMSE}_{\mathrm{VI}} = 0.109$), Posterior coverage is good for `metabeta` ($\mathrm{CE}_{\mathrm{MB}} = 0.007$), whereas HMC's marginal posterior for the variance parameters tend to be slightly too wide ($\mathrm{CE}_{\mathrm{HMC}} = 0.094$) and the ones of VI too narrow ($\mathrm{CE}_{\mathrm{VI}} = -0.053$). The median posterior fits are in the same neighborhood ($\mathrm{NLL}_{\mathrm{MB}} = 805.1$ vs. $\mathrm{NLL}_{\mathrm{HMC}} = 829.0$ vs. $\mathrm{NLL}_{\mathrm{VI}} = 802.6$) and posterior fits are highly correlated ($r_{\mathrm{MB,HMC}} = 0.940$, $r_{\mathrm{MB,VI}} = 0.940$, $r_{\mathrm{HMC,VI}} = 0.999$). Overall, all models perform excellently on the toy problem and make very similar predictions. This shows that the pipeline is in principle correctly specified for each approach.

### 3.2 REAL PREDICTORS, SIMULATED PARAMETERS

To get a better estimate for model performance under more realistic conditions, we sample predictors ($\mathbf{X}$) from a large set of real datasets and combine them with synthetically sampled regression parameters ($\vartheta$) to produce regression outcomes ($\mathbf{y}$). The test sets were constructed using the same simulation pipeline as the training sets, but with different random seeds. Please find the details of this approach in Appendix A. This approach of testing on semi-synthetic datsets has the following considerable benefits over evaluation on purely real data: (1) The regression models are always correctly specified, (2) we know the ground truth parameters and can thus evaluate parameter recovery and coverage, (3) we can compare the results to in-distribution test data and gauge how well the model transfers to realistic predictors (Lueckmann et al., 2021; Ward et al., 2022).

Table 1 shows model performance for hierarchical regression problems with increasing numbers of fixed and random parameters. Recovery and coverage per parameter type are visualized inFigure 2 for $d = 5, q = 2$. Median model fits of `metabeta` and HMC are very similar: While HMC generally explains the outcomes better, our model outperforms VI in this regard. Over the test batch, the averge model fits are highly correlated between `metabeta` and HMC ($r = 0.94$), indicating overall high agreement between both methods. Similarly, parameter recovery is best for HMC, but its advantage over `metabeta` is very small. VI performs similarly well for the two smaller problems but struggles more with the latter two. Posterior coverage is generally best for `metabeta`, as its CE is closest

to 0 in all cases. In comparison, HMC has unstable coverage and VI tends towards too narrow posteriors. Lastly, the median run time for a single dataset is almost instantaneous for metabeta, strongly outperforming both other methods. Overall, our model appears to have comparably stable performance to HMC and outperforms VI, which marks metabeta as a strong alternative to HMC if practitioners are willing to accept a minor reduction in accuracy for a substantial boost in speed.

**Table 1:** Performance evaluation for metabeta, HMC and VI on semi-synthetic test sets with $d$ fixed effects and $q$ random effects. The test sets contain real predictors $\mathbf{X}$ and simulated regression parameters. The evaluation metrics are negative log-likelihood (NLL $= -\log p(\mathbf{y}|\hat{\boldsymbol{\vartheta}})$), Pearson's correlation-coefficient $r$, root mean squared error RMSE, and coverage error CE$(\alpha)$ averaged over $\alpha \in \{0.05, 0.1, 0.2, 0.32, 0.5\}$, as well as median runtimes in seconds. Bold formatting indicates better performance.

| $d$ | $q$ | model | NLL | $r$ | RMSE | CE | seconds |
|---|---|---|---|---|---|---|---|
| 3 | 1 | metabeta | 456.1 | **0.987** | 0.059 | **0.028** | **0.01** |
| | | HMC | **423.7** | **0.987** | **0.058** | 0.046 | 12.48 |
| | | VI | 528.7 | 0.983 | 0.089 | -0.161 | 4.49 |
| 5 | 2 | metabeta | 355.5 | 0.966 | 0.079 | **0.014** | **0.01** |
| | | HMC | **351.7** | **0.976** | **0.067** | 0.037 | 13.68 |
| | | VI | 479.8 | 0.967 | 0.092 | -0.224 | 9.41 |
| 8 | 3 | metabeta | 438.3 | **0.977** | **0.048** | **-0.037** | **0.01** |
| | | HMC | **417.8** | 0.964 | 0.092 | -0.138 | 15.59 |
| | | VI | 642.2 | 0.883 | 0.405 | -0.347 | 12.75 |
| 12 | 4 | metabeta | 534.1 | 0.938 | 0.106 | **0.040** | **0.01** |
| | | HMC | **504.7** | **0.945** | **0.099** | -0.205 | 36.96 |
| | | VI | 757.2 | 0.849 | 0.511 | -0.398 | 21.58 |

**Table 2:** Performance evaluation for metabeta, HMC and VI on various subsets of real hierarchical datasets with unknown regression parameters. Performance is evaluated on in-sample posterior accuracy as measured by median negative log-likelihood, $-\log p(\mathbf{y}|\hat{\boldsymbol{\vartheta}})$. Bold formatting indicates better performance.

| | Sleep | Gcsemv | Exam | Math | Titanic | Schooling | News |
|---|---|---|---|---|---|---|---|
| metabeta | 109.4 | 390.9 | 784.8 | 883.2 | 810.8 | 738.2 | 540.3 |
| HMC | 115.2 | **389.2** | **773.5** | **856.4** | 788.7 | **640.9** | 400.0 |
| VI | **105.3** | 403.3 | 782.2 | 869.1 | **787.6** | 661.9 | **399.3** |

## 3.3 REAL DATASETS

We gathered 7 canonical datasets that are often used for demonstration purposes of mixed-effects regression and ran each model on multiple subsets thereof. No parameters are simulated for these test sets and we use the default prior specification of Bambi for posterior estimation. To gauge model fits, we compare in-sample posterior accuracy with the same methods as above. Table 2 lists model performance for all datasets. Overall, the median NLL of our model is very similar to that of HMC and VI, which also shows in the average correlation over batches, ($r_{\text{MB,HMC}} = 0.880$, $r_{\text{MB,VI}} = 0.876$, $r_{\text{HMC,VI}} = 0.951$). This indicates general agreement posterior inference, even on out-of-distribution data with likely miss-specified priors and model structure.

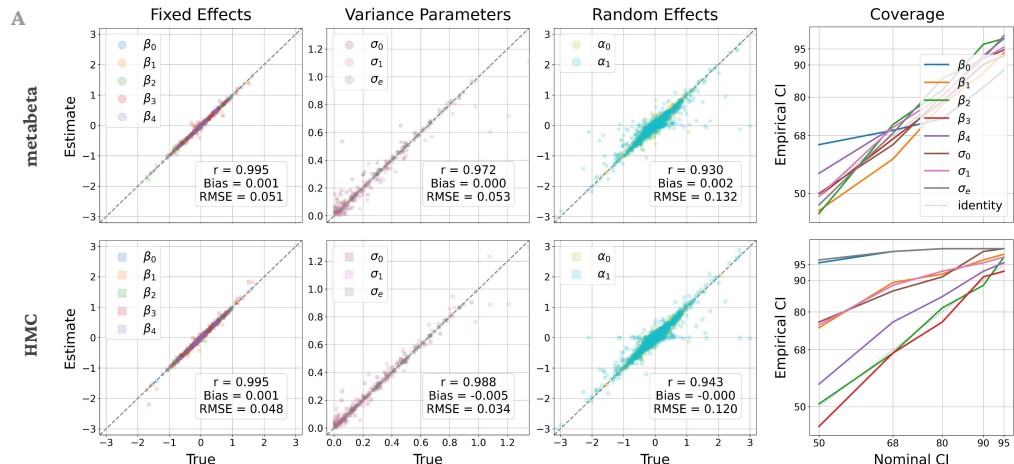

**Figure 2:** Model performance for $d = 5$ and $q = 2$. Results for other regression problems are depicted in Appendix G. (A) *Parameter Recovery.* Our model reaches similar performance to HMC in terms of r, bias and RMSE for all parameter types. (B) *Coverage.* Our model's posterior credible intervals are on average more faithfully tuned, whereas the HMC posteriors tend to be unnecessarily broad.

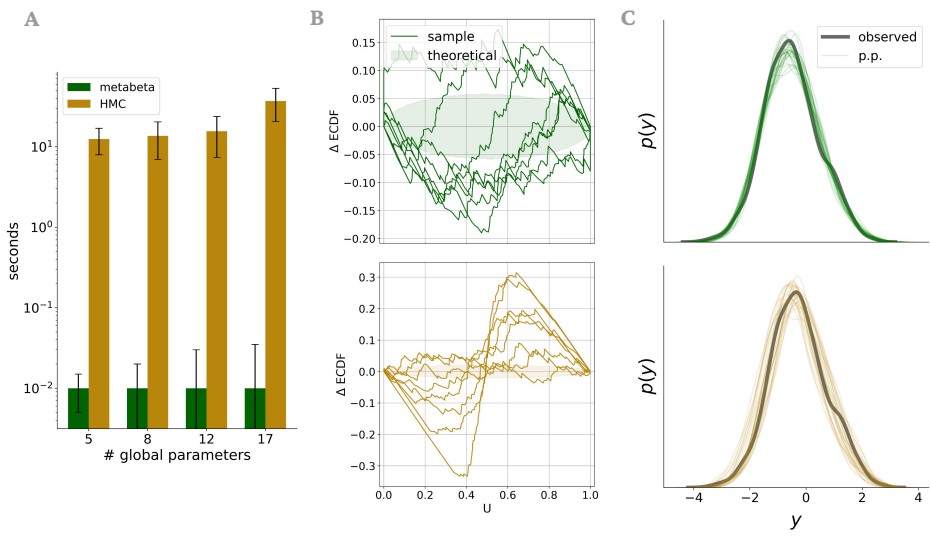

**Figure 3:** (A) Median runtimes per dataset for `metabeta` and `HMC`. Error bars symbolize standard deviations. Our model is orders of magnitudes faster than MCMC. (B) Simulation based calibration (SBC) comparing the sample distributions of parameter rank statistics against the uniform distribution. Plots are stacked for multiple parameters. Calibration is closer to the optimum for our model. (C) Example posterior predictive distributions based on samples from both models.

## 4 DISCUSSION

In this paper we present `metabeta`, a probabilistic transformer-based model that performs efficient approximate Bayesian inference for mixed-effects regression. We trained `metabeta` on simulated datasets with varying ranges for predictors, regression parameters, and outcomes. Most importantly, these datasets incorporate varying priors and we condition the model outputs on them, which not only amortizes the high computational costs encountered when using MCMC for parameter estimation, but also generalizes previous neural posterior estimation (NPE) techniques that are trained on a fixed prior. We show that our model has favorable and robust performance on in-distribution and out-of-distribution test sets, based on real hierarchical datasets practitioners care about. In each experiment, we compare the results of our model with Hamiltonian Monte Carlo (HMC) and variational inference

(VI), the gold-standard methods for Bayesian mixed-effects regression, and show that `metabeta` generally approaches HMC in model fit, accuracy, and fidelity of credible intervals and outperforms VI in most. Most importantly, it does that at a small fraction of the time required for parameter estimation with conventional methods.

The high speed and explicit incorporation of priors opens new avenues for Bayesian mixed-effects regression: Analysts can now specify multiple priors simultaneously and check how robust the model posteriors are to varying a priori assumptions. Furthermore, it is straightforward to extend our model to a mixture of experts by passing the same dataset multiple times with different permutations of the design matrix columns, and then aggregating the resulting back-permuted posterior samples (Hollmann et al., 2025).

## 4.1 Limitations and Outlook

Our choice of model architecture trades of posterior expressivity for computation speed: Other normalizing flow methods like Neural Spline Flows (Durkan et al., 2019), Flow Matching (Wildberger et al., 2023), Conditional Diffusions (Chen et al., 2025; Reuter et al., 2025) or TarFlow (Zhai et al., 2025) offer more flexible posterior shapes, but posterior sampling is considerably more expensive than for Affine Coupling Flows, often involving numerical integration or solving a stochastic differential equation. The relative simplicity of affine coupling posteriors can be seen as implicit regularization, preventing overly irregular quantification of regression parameter uncertainty.

Each trained version of metabeta is currently tailored to the size of the regression problem in terms of the number of fixed effects ($d$) and the number of random effects ($q$). The GitHub repository provides pretrained versions of `metabeta` for several relevant parameter combinations. Together this collection of models acts like a single pretrained model, as each size can be pulled quickly from the repo for immediate deployment. That is, from the practitioners perspective it makes no difference if there is a single or multiple pretrained models for different regression problem sizes.

Currently, the prior families are fixed. The parameters of the priors are concatenated to the summary vector $\mathbf{s}$ before being passed to the MLPs inside the normalizing flow. This approach could be generalized to varying prior families, whose identity can be embedded and simply added to the summary vector. We plan to eventually extend `metabeta` to even more flexible prior specification. Similarly, our framework is currently specialized on Bayesian linear mixed effects regression, but the required steps to generalized mixed-effects models are in parts small: Data simulation would require an additional response function around the linear term. The response function type could be passed to the model along with the priors. Extending importance sampling for non-linear cases is non-trivial, however. Finally, hierarchical NPE is well suited for mixed-effects regression with one grouping factor: Multiple parallel grouping factors would require non-trivial extensions to dataset summarization and integration of multiple summaries. However, it is conceptually straightforward to extend the framework to multiple nested grouping factors (e.g. schools and classrooms within schools). Overall, these extensions are worthwhile avenues for future developments.

## 4.2 Conclusion

Our model brings Bayesian mixed-effects regression closer to practical usability in real-world applications. In its current form, it already enables rapid prototyping – practitioners can quickly test different model specifications and validate findings using conventional tools if needed. Our analyses highlight that `metabeta` is immediately applicable to such use cases. Looking ahead, we envision scaling our model to larger and non-linear regression problems. This would open the door to entirely new applications of Bayesian mixed-effects regression that are currently out of reach.

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

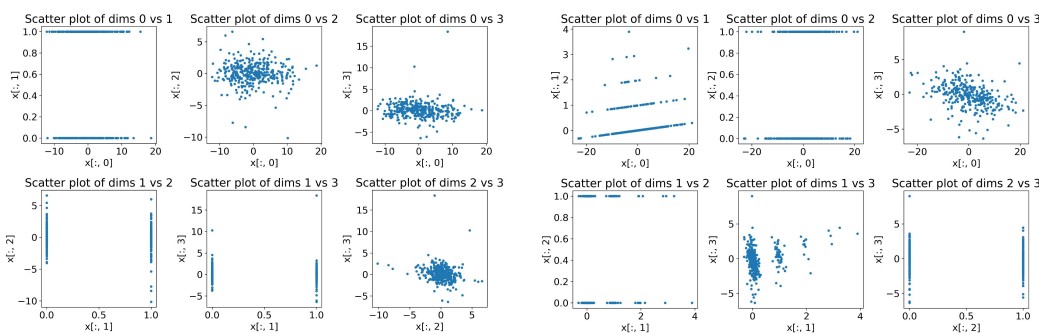

**Figure 4:** Scatter plots of sampled synthetic predictors for two datasets.

## A    DATASET SIMULATION

All simulated or sampled datasets have $5 \leq m \leq 30$ groups and each group has $10 \leq n_i \leq 70$ observations, making $n$ range from 50 to 2100.

### A.1    SYNTHETIC PREDICTORS

We sample $n_i$ observations of predictor $j$, $\mathbf{x_{ij}}$, from the following distributions: Normal, Student-$t$, continuous uniform, log-Normal, Bernoulli, negative binomial, and scaled Beta. All distributions have varying parameters and include random truncations. Correlation is induced by sampling $\mathbf{LL}^\top = \mathbf{R} \sim \mathrm{LKJ}(10)$ and multiplying $\mathbf{L}$ with the design matrix $\mathbf{X}$ (Lewandowski et al., 2009). For binary variables, we induce correlation with another variable using the following approach:

---

**Algorithm 1:** Sample correlated binary variable

**Data:** $\mathbf{x} \in \mathbb{R}^n$, $r \in (-1, 1)$
**Result:** $\mathbf{z} \in \{0, 1\}^n$
$\mathbf{y} \sim \mathcal{N}_n(0, 1)$;
$\mathbf{y} \leftarrow r \cdot \mathbf{x} + (1 - r^2)^{\frac{1}{2}} \cdot \mathbf{y}$;
$\mathbf{p} \leftarrow (1 + e^{-\mathbf{y}})^{-1}$;
$\mathbf{z} \sim \mathrm{Bernoulli}(\mathbf{p})$;

---

An example of generated training data is visualized in Figure 8.

### A.2    REAL PREDICTORS

We use 271 real datasets from the PMLB (Romano et al., 2021) and SRM (Lichtenberg & Şimşek, 2017) benchmarks as additional sources for realistic predictors, and preserve hierarchical grouping structure when present. Existence of grouping structure is automatically checked by checking every non-continuous predictor for its number of unique values, as well as their spread. When such grouping factors are present, data is separately sampled per group, otherwise groups are randomly assigned. To further increase variability, we pass the sampled real data through Stochastic Gradient Langevin Dynamics (SGLD, Welling & Teh, 2011; Raginsky et al., 2017; Ma et al., 2024). This generates structurally equivalent data instead of just using subsets. The training sets receive a mix of synthetic and emulated predictors, the test sets receive only real data subsets. The in-distribution test sets rely on samples from the 271 datasets, the out-of-distribution sets rely on 7 additional hierarchical datasets not used in the training data.

### A.3 PRIORS AND RESCALING

Priors for parameters are sampled using the following approach:

---

**Algorithm 2:** Sample priors

---

**Data:** $b \in \mathbb{N}, d \in \mathbb{N}, q \in \mathbb{N}$
**Result:** $\boldsymbol{\nu}_\beta \in \mathbb{R}^{b \times d}, \boldsymbol{\tau}_\beta \in \mathbb{R}^{b \times d}, \boldsymbol{\tau}_\sigma \in \mathbb{R}^{b \times q}, \boldsymbol{\tau}_\varepsilon \in \mathbb{R}^b$
$\boldsymbol{\nu}_\beta \sim \mathcal{U}_{b \times d}(-3, 3);$
$\boldsymbol{\tau}_\beta \sim \mathcal{U}_{b \times d}(0.01, 3);$
$\boldsymbol{\tau}_\sigma \sim \mathcal{U}_{b \times q}(0.01, 3);$
$\boldsymbol{\tau}_\varepsilon \sim \mathcal{U}_{b \times 1}(0.01, 3);$

---

In a first forward pass, the standardized predictors and sampled parameters are projected to $\mathbf{y}$. Then all parameters (and their corresponding prior parameters) are rescaled such that $\mathbb{V}(\mathbf{y}) = 1$. This is without loss of generality, as posterior samples can trivially be brought back to the original scale of $\mathbf{y}$ by rescaling (see below). However, the advantage of this approach is that this covers a very wide range of potentially observable combinations of $\mathbf{X}$ and $\boldsymbol{\vartheta}$.

## B STANDARDIZATION

Before entering the neural model, all observable data is normalized to zero mean and unit standard deviation over groups and observations. To keep the dependence structure intact, we also analytically standardize the regression parameters during training and un-standardize them after sampling, using the following equalities:

$$\beta_k^* = \beta_k \frac{\sigma_{x_k}}{\sigma_y}$$

$$\alpha_{ik}^* = \alpha_{ik} \frac{\sigma_{z_k}}{\sigma_y} \sim \mathcal{N}\left(0, \sigma_k^{*2}\right)$$

$$\sigma_k^{*2} = \sigma_k^2 \frac{\sigma_{z_k}^2}{\sigma_y^2}, \quad \sigma_\varepsilon^{*2} = \frac{\sigma_\varepsilon^2}{\sigma_y^2}$$

where $\sigma_{x_k}$ resp. $\sigma_y$ are the $k$th predictor's resp. the outcome's standard deviation, and $\beta_k^*$ is the $k$th slope after z-standardizing predictors and outcomes. The intercepts require special care:

$$\beta_0^* = \frac{\beta_0 + \sum_{k=1}^d \mu_{x_k} \beta_k - \mu_y}{\sigma_y}$$

$$\alpha_{i0}^* = \frac{\alpha_{i0} + \sum_{k=1}^q \mu_{z_k} \alpha_{ik}}{\sigma_y} = \frac{\sum_{k=0}^q \mu_{z_k} \alpha_{ik}}{\sigma_y} \sim \mathcal{N}\left(0, \sigma_0^{*2}\right),$$

where $\mu_{x_k}$ is the mean of the $k$th predictor over all observations. Due to the sum term in the latter,

$$\sigma_0^{*2} = \mathbb{V}\left(\alpha_{i0}\right) + \mathbb{V}\left(\sum_{k=1}^q \mu_{z_k} \alpha_{ik}\right) + 2 \cdot \text{Cov}\left(\alpha_{i0}, \sum_{k=1}^q \mu_{z_k} \alpha_{ik}\right),$$

which is equivalent to summing up the covariance matrix of the random vector $\boldsymbol{\mu}_z \odot \boldsymbol{\alpha}_i$.

*Proof*:

$$y_{ij}^* = \frac{y_{ij} - \mu_y}{\sigma_y}$$

$$= \frac{1}{\sigma_y} \left( \beta_0 + \sum_{k=1}^{d} x_{ijk}\beta_k + \alpha_{i0} + \sum_{k=1}^{q} z_{ijk}\alpha_{ik} + \varepsilon_{ij} - \mu_y \right)$$

$$\stackrel{!}{=} \beta_0^* + \sum_{k=1}^{d} x_{ijk}^*\beta_k^* + \alpha_{i0}^* + \sum_{k=1}^{q} z_{ijk}^*\alpha_{ik}^* + \varepsilon_{ij}^*$$

$$= \beta_0^* + \sum_{k=1}^{d} \left( \frac{x_{ijk} - \mu_{x_k}}{\sigma_{x_k}} \right) \beta_k^* + \alpha_{i0}^* + \sum_{k=1}^{q} \left( \frac{z_{ijk} - \mu_{z_k}}{\sigma_{z_k}} \right) \alpha_{ik}^* + \varepsilon_{ij}^*$$

$$= \beta_0^* + \sum_{k=1}^{d} \left( \frac{x_{ijk} - \mu_{x_k}}{\sigma_{x_k}} \right) \left( \beta_k \frac{\sigma_{x_k}}{\sigma_y} \right) + \alpha_{i0}^* + \sum_{k=1}^{q} \left( \frac{z_{ijk} - \mu_{z_k}}{\sigma_{z_k}} \right) \left( \alpha_{ik} \frac{\sigma_{z_k}}{\sigma_y} \right) + \frac{\varepsilon_{ij}}{\sigma_y}$$

$$= \beta_0^* - \sum_{k=1}^{d} \left( \frac{\mu_{x_k}\beta_k}{\sigma_y} \right) + \sum_{k=1}^{d} \left( \frac{x_{ijk}\beta_k}{\sigma_y} \right) + \alpha_{i0}^* - \sum_{k=1}^{q} \left( \frac{\mu_{z_k}\alpha_{ik}}{\sigma_y} \right) + \sum_{k=1}^{q} \left( \frac{z_{ijk}\alpha_{ik}}{\sigma_y} \right) + \frac{\varepsilon_{ij}}{\sigma_y}$$

$$= \frac{\beta_0 - \mu_y}{\sigma_y} + \sum_{k=1}^{d} \left( \frac{x_{ijk}\beta_k}{\sigma_y} \right) + \frac{\alpha_{i0}}{\sigma_y} + \sum_{k=1}^{q} \left( \frac{z_{ijk}\alpha_{ik}}{\sigma_y} \right) + \frac{\varepsilon_{ij}}{\sigma_y}$$

$\square$

The distributions of the standardized random effects and noise follow from the scaling of normal random variables, and the variance of the random intercept follows from the variance of sums of random variables (Wasserman, 2010).

## C  DATA REPRESENTATION AND EMBEDDING

Group-membership is represented implicitly by a separate tensor dimension, e.g. $\mathbf{X}$ has the shape $(batch, m, n, d)$. For `PyTorch` dataloader compatiblity, all tensors are zero-padded and corresponding masks are stored. To spread the learning signal evenly across the network, all slope-related variables are randomly permuted separately per regression dataset, using the same permutation for $\mathbf{X}, \mathbf{Z}, \boldsymbol{\beta}, \boldsymbol{b}_i$, and $\mathbf{S}$.

Observable data is concatenated along the last dimension to $\mathbf{D} = [\mathbf{y}, \mathbf{X}, \mathbf{Z}]$, and linearly projected to a higher-dimensional space (e.g. 128 dimensions). Since mixed-effects regression must be permutation invariant (wrt. to groups and observations per group), no positional encoding or explicit group identity information is passed as input, and instead group identity is represented implicitly by a separate tensor dimension, e.g. $\mathbf{X}$ has the shape $(batch, m, n, d)$.

## D  POSTERIOR FACTORIZATION

Let the joint distribution over all regression parameters and the data be

$$p(\boldsymbol{\vartheta}, \boldsymbol{\alpha}, \mathbf{D}),$$

where $\boldsymbol{\alpha} = \{\boldsymbol{\alpha}_i\}_{i=1,\dots,m}$ and $\mathbf{D} = \{\mathbf{D}_i\}_{i=1,\dots,m}$.

We can write the joint posterior as

$$\frac{p(\boldsymbol{\vartheta}, \boldsymbol{\alpha}, \mathbf{D})}{p(\mathbf{D})} = p(\boldsymbol{\vartheta}, \boldsymbol{\alpha} \,|\, \mathbf{D}) = p(\boldsymbol{\vartheta} \,|\, \mathbf{D})\, p(\boldsymbol{\alpha} \,|\, \boldsymbol{\vartheta}, \mathbf{D}) = p(\boldsymbol{\vartheta} \,|\, \mathbf{D}) \prod_{i=1}^{m} p(\boldsymbol{\alpha}_i \,|\, \boldsymbol{\vartheta}, \mathbf{D}_i),$$

where we use the conditional independence of the local parameters in the last step. This translates naturally to the loss calculation:

$$\ell = \ell_{\Pi_g, \Sigma_g} + \sum_{i=1}^{m} \ell_{\Pi_l, \Sigma_l}^{(i)} \propto -\mathbb{E}_{\boldsymbol{\vartheta}, \boldsymbol{\alpha}, \mathbf{D}} \left[ \log p_{\Pi_g} \left( \boldsymbol{\vartheta} \mid f_{\Sigma_g}(f_{\Sigma_l}(\mathbf{D})) \right) + \sum_{i=1}^{m} \log p_{\Pi_l} \left( \boldsymbol{\alpha}_i \mid \boldsymbol{\vartheta}, f_{\Sigma_l}(\mathbf{D}_i) \right) \right].$$

Similar derivations can be found in Heinrich et al. (2023) and Habermann et al. (2024).

## E    ALTERNATING IMPORTANCE SAMPLING

For numerical stability, we compute

1. $\log w_i \leftarrow \log p(\mathbf{D}|\boldsymbol{\vartheta}_i) + \log p(\boldsymbol{\vartheta}_i) - \log q(\boldsymbol{\vartheta}_i|\mathbf{D})$
2. $\log w_i \leftarrow \min(\log w_i, \log w^\dagger)$, where $\log w^\dagger$ is the 98th percentile over $i$
3. $w_i \leftarrow \exp(\log w_i - \max_j \log w_j)$, such that $w_i \leq 1$ for all $i$
4. $\tilde{w}_i \leftarrow \frac{w_i}{\frac{1}{s}\sum_{i=1}^{s} w_i}$ such that $\sum_{i=1}^{s} \tilde{w}_i = s$.

Since we have two approximate posteriors (one for the global parameters, one for the random effects), we have two sets of samples which require separate importance weights (IW). For the global parameters posterior, the numerator can either use the *marginal* likelihood,

$$p(\mathbf{D}|\boldsymbol{\vartheta})p(\boldsymbol{\vartheta}) = \prod_{i=1}^{m} p(\mathbf{y}_i|\mathbf{X}_i, \boldsymbol{\beta}, \boldsymbol{\sigma}_\alpha^2, \sigma_\varepsilon^2)p(\boldsymbol{\beta})p(\boldsymbol{\sigma}_\alpha^2)p(\sigma_\varepsilon^2),$$

or the *conditional* likelihood,

$$p(\mathbf{D}|\boldsymbol{\vartheta})p(\boldsymbol{\vartheta}) = \prod_{i=1}^{m} p(\mathbf{y}_i|\mathbf{X}_i, \boldsymbol{\alpha}_i, \boldsymbol{\beta}, \sigma_\varepsilon^2)p(\boldsymbol{\alpha}_i|\boldsymbol{\sigma}_\alpha^2)p(\boldsymbol{\sigma}_\alpha^2)p(\boldsymbol{\beta})p(\sigma_\varepsilon^2).$$

The marginal likelihood may seem more appropriate, because the global posterior does not receive any explicit information about the random effects, i.e. it is not conditioned on them. However, calculating the marginal likelihood is inefficient, as it requires a matrix inversion for each sample. Empirically, parameters recovery also suffers from using marginal likelihood IW. Instead, we plug in the posterior mean of the random effects for the conditional likelihood IW. The IW for the random effects posterior is calculated accordingly, this time using the importance-weighted means of the global parameters. We alternate the two steps 3 times, starting with the local samples.

## F    FIT DIAGNOSTICS

To do

# G    RESULT FIGURES

Descriptors are the same as in Figure 2 and Figure 3B.

## G.1    TOY EXAMPLE

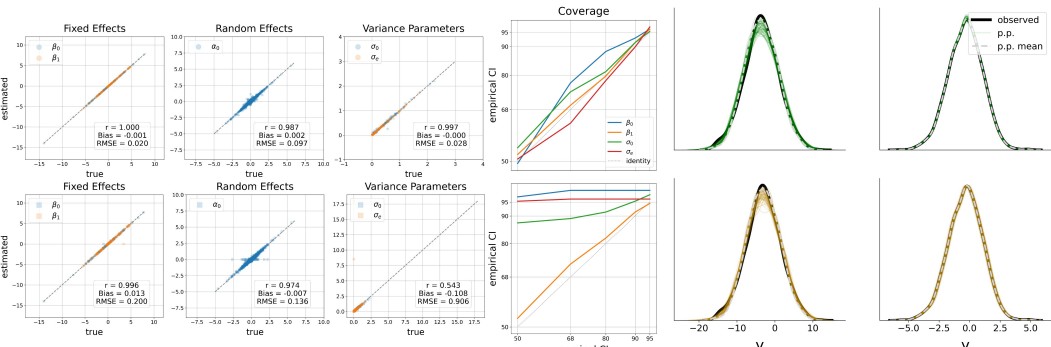

**Figure 5:** Results based on the toy example.

## G.2    EXAM

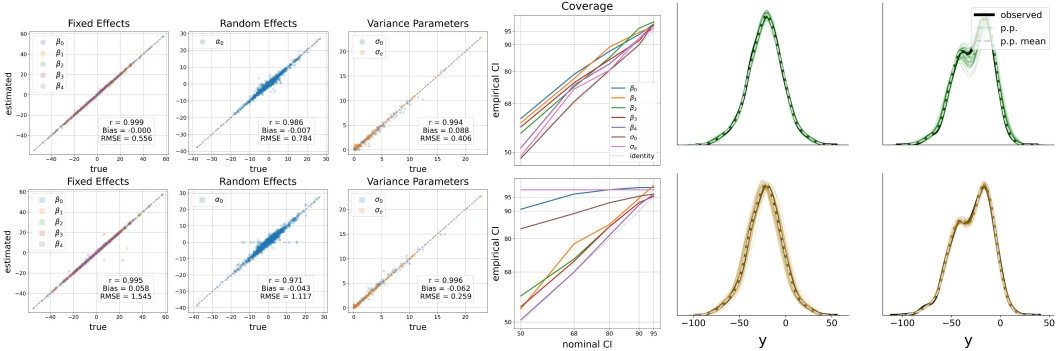

**Figure 6:** Results based on `Exam`.

## G.3 GCSEMV

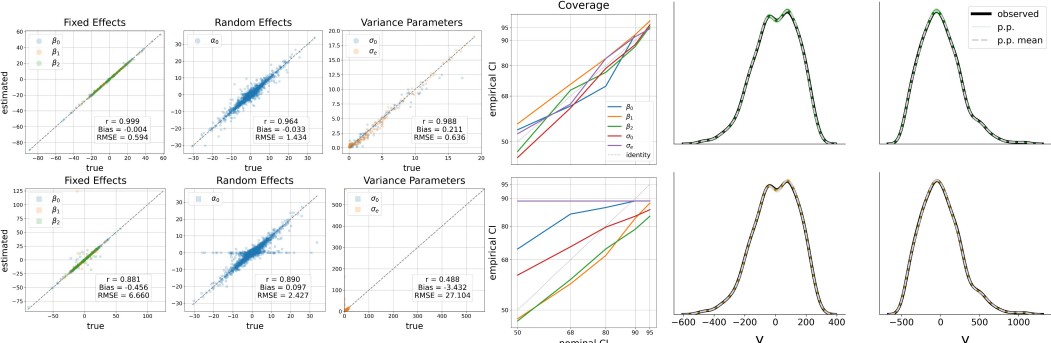

**Figure 7:** Results based on `Gcsemv`.

## G.4 SLEEPSTUDY

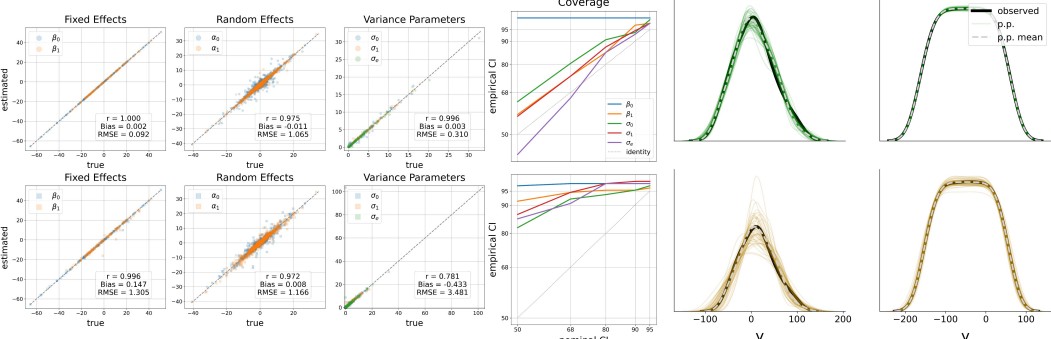

**Figure 8:** Results based on `Sleepstudy`.

