# OpenReview forum: "metabeta - A fast neural model for Bayesian mixed-effects regression"
_ICLR.cc/2026/Conference — Submitted to ICLR 2026_

### Official Review · Reviewer_4rdr · 2025-10-20

**Soundness:** 2
**Presentation:** 3
**Contribution:** 4
**Rating:** 6
**Confidence:** 5

**Summary:**

The paper presents a simulation-based inference approach for Bayesian mixed-effects regression called metabeta. Hierarchical models are an important and widely used class of models, and efficient inference for them remains an active research topic. The authors train a single model that supports multiple priors, instead of training one model per prior as in previous work, and they employ a transformer-based architecture. The approach aims to achieve MCMC-level accuracy with greatly reduced inference time. The paper is clearly written, well-motivated, and addresses an interesting and timely problem in Bayesian inference for hierarchical models. It builds upon recent advances such as those by Habermann et al. and  Hollmann et al., extending neural posterior estimation to a broader mixed-effects setting with multiple priors. This extension is both conceptually interesting and practically meaningful.

**Strengths:**

This paper addresses an important and timely problem: Bayesian inference for hierarchical models. The proposed metabeta framework is interesting, combining transformer-based neural posterior estimation with amortization across multiple priors. The approach shows promise and could meaningfully reduce inference cost in mixed-effects regression. Strengths are
- Novel and well-motivated application of neural posterior estimation to mixed-effects regression, and effectively leveraging the available likelihood for post-hoc refinement.
- The integration of multiple priors into a single amortized inference model represents a significant step toward general-purpose Bayesian inference.
- Comprehensive benchmark suite covering both in-distribution and out-of-distribution data, which supports the claims of generalisation and robustness.
- Clear presentation and placement within the existing literature, helping the reader understand the contributions of the approach.

**Weaknesses:**

HMC diagnostics are not reported and convergence is doubtful

The paper reports “divergence and strong outliers” (l.129) in HMC runs but does not provide diagnostics and the authors do not define what constitutes “outliers” (l. 129). Moreover, they do not report effective sample sizes, divergence counts or a similar convergence diagnostics. Unconverged chains should be excluded and metrics computed on all converged runs. The current MAD-based chain selection may bias variance estimates, potentially causing the selected chain of HMC to underestimate posterior variance. This should be justified or replaced with a more standard convergence diagnostic. If convergence issues persist, HMC might need a longer warm-up or more iterations.

More concretely:
Figure 1D shows a discrepancy between HMC and metabeta. Given that HMC has asymptotic convergence guarantees, and that HMC uses the “true priors and generative model” (l.128) , this raises questions about whether metabeta is overconfident (e.g., for $\beta_0$) or whether HMC was improperly tuned. If the latter, longer runs or a different parameterization may be required. Please clarify the setup and report diagnostics. To support the claim that both pipelines are correctly specified, including at least one simple case with a known analytical posterior could be helpful to demonstrate that HMC converges and metabeta matches it. Even when HMC turns out to perform better, the amortisation part is still very valuable. Interesting would also be to compare the methods, when the prior is misspecified.

A quantification of the differences between HMC and metabeta in the posterior predictive distributions would further strengthen the results.

Minor comments
- In Section 2.1, $S$ is undefined (l.95).
- “we simulate hierarchically structured datasets using PyTorch“ (l.110), what is the benefit of using PyTorch here for simulation?
- Figure 1 needs to be improved in terms of overall readability: missing labels (panel C), unreadable fonts, neural networks are badly visible, and too small text in subplots. Make clear that D is only inference time, not training time.
- Font sizes in all main figures are too small. In general, the resolution of the figures should be improved.
- The importance sampling refinement is a nice idea. Showing results without this step to isolate its contribution would further improve the paper.
- The code is not provided, preventing verification of the claim that open-source software and pretrained models are or will become available. Please check out https://anonymous.4open.science/.

**Questions:**

The authors claim that prior work “at best nullifies the runtime advantage of NPE” (l. 68). How does this compare to the authors’ method? In particular there are serval situations, which are not discussed enough in the opinion of the reviewer, such as:

- How can missing data be handled? Many hierarchical datasets contain incomplete observations, and prior work uses masking to circumvent this issue (e.g., “All-in-one simulation-based inference“ by Gloeckler et al.).

- The paper claims full amortization, yet separate models are trained for different parameter dimensionalities. This weakens the claim of applicability for practitioners if the needed parameter dimensionality is not available. The authors should clarify early in the manuscript that amortization holds only for fixed dimensionality $d$ and group structure $q$ and that multiple models are trained. Also, here they could connect to recent work, such as "Compositional amortized inference for large-scale hierarchical Bayesian models" by Arruda et al.

- How are priors incorporated as inputs: through direct parameterization or learned embeddings? What about priors which are not part of the training data? This should be clarified in the main maunscript and potentially the relation to Whittle et al. “Distribution Transformers: Fast Approximate Bayesian Inference with On-The-Fly Prior Adaptation“ discussed, which seems closely related.

Please report computational cost for training metabeta, including total training time for the different models and resource usage. This is essential for assessing amortization trade-offs.

---

> ### Author Response · Authors · 2025-11-21
>
> We strongly thank the reviewer for the rewarding in-depth review and encouragement. We are happy they like the presentation and deem the contribution significant and comprehensive.
>
> > HMC diagnostics are not reported and convergence is doubtful [...]
>
> The criticism is warranted and we have greatly improved the HMC fits by using NUTS instead of vanilla HMC, including non-centered parameterization of the random effects, and taking samples from four chains. Additionally, we save fit statistics like divergence counts, effective sample size and R-hat statistics and will add them to the SI.
>
> We have implemented three different setups: (1) one that matches the structure of the data generating process (distinguishing between fixed and random effects), (2) one with hierarchical parameterization that extends the official example by PyMC (distinguishing between mixed effects and pure fixed effects without additional random offsets), and (3) a Bambi-based setup with either default priors (specified by Bambi), true priors, and a hybrid version where the fixed effects priors are specified by Bambi and the others match the true priors. Hybrid versions of the two PyMC-based setups are included too, automatically specifying the priors for the fixed effects based on the moments of X and y (exactly matched with Bambi’s procedure). The table below compares each variant on a toy test set with 256 datasets in terms of parameter recovery (r, RMSE), average posterior width (SD) and average R-hat. The best values are marked bold.
>
> | Method | Parameter | r | RMSE | SD | R-hat |
> |-|-|-|-|-|-|
> | **Bambi: default** | fixed effects | **1.000** | 0.072 | 0.258 | 1.007 |
> | | random effects | **0.986** | 0.309 | 0.279 | 1.006 |
> | | variances | 0.994 | 0.141 | 0.219 | 1.006 |
> | **Bambi: true** | fixed effects | 0.999 | 0.110 | **0.231** | **1.003** |
> | | random effects | 0.984 | 0.327 | 0.258 | **1.004** |
> | | variances | **0.997** | 0.078 | 0.179 | 1.005 |
> | **Bambi: hybrid** | fixed effects | **1.000** | **0.062** | 0.237 | 1.004 |
> | | random effects | **0.986** | **0.307** | 0.263 | 1.005 |
> | | variances | **0.997** | **0.077** | **0.177** | 1.007 |
> | **PyMC: matched-true** | fixed effects | 0.999 | 0.113 | **0.230** | 1.004 |
> | | random effects | 0.984 | 0.327 | **0.257** | **1.004** |
> | | variances | 0.996 | 0.079 | 0.178 | **1.004** |
> | **PyMC: matched-hybrid** | fixed effects | **1.000** | **0.062** | 0.239 | 1.004 |
> | | random effects | **0.986** | **0.307** | 0.265 | 1.005 |
> | | variances | **0.997** | 0.078 | **0.177** | 1.006 |
> | **PyMC: hierarchical-true** | fixed effects | 0.999 | 0.119 | 0.231 | 1.005
> ||random effects | 0.984 | 0.330 | 0.258 | 1.005
> ||variances | 0.996 | 0.085 | 0.180 | **1.004**
> | **PyMC: hierarchical-hybrid** | fixed effects | **1.000** | 0.064 | 0.239 | 1.004 |
> | | random effects | **0.986** | **0.307** | 0.265 | 1.005 |
> | | variances | 0.996 | 0.079 | 0.178 | **1.004** |
>
> This marks the hybrid versions of Bambi and PyMC (matched) as the top contenders. The next table compares the divergence behavior (mean number of divergent samples, frequency of any divergent sample occurring) over variants:
>
> | Method | mean | frequency |
> |-|-|-|
> | **Bambi: default** | 2.394 | 0.254
> | **Bambi: true** | **0.971** | 0.193
> | **Bambi: hybrid** | 1.473 | 0.187
> | **PyMC: matched-true** | 1.576 | 0.178
> | **PyMC: matched-hybrid** | 2.185 | **0.165**
> | **PyMC: hierarchical-true** | 1.462 | 0.175
> | **PyMC: hierarchical-hybrid** | 1.699 | 0.186
>
> This suggests, the PyMC-hybrid configuration has the rarest divergence occurrence (but when it diverges, it has a relatively high number of divergent samples, with 2 outlier datasets out of 256). On the other hand, hybrid Bambi has a relatively higher divergence probability (but when it diverges, it produces relatively fewer divergent samples). We believe this overall favors PyMC-matched-hybrid. For fair comparisons, datasets with high divergence will be removed from the test set. **We will rerun all experiments with the improved implementation of HMC and will also report diagnostics in the SI until the end of the rebuttal phase.**
>
>
> > A quantification of the differences between HMC and metabeta in the posterior predictive distributions would further strengthen the results.
>
> Thank you for the suggestion. We will include posterior predictive accuracy as an evaluation metric for model comparisons in the reruns until the end of the rebuttals.
>
> > In Section 2.1, S is undefined (l.95).
>
> Thanks for the pointer, we will add the following footnote:
>
> *$\mathbf S$ is generally any symmetric positive-definite matrix in $\mathbb R^{q \times q}$. To start with a proof of concept, we use the simpler but already useful case of diagonal $\mathbf S$, i.e. each random effect can have its own variance, but is sampled independently.*

---

> ### Author Response · Authors · 2025-11-22
>
> > “we simulate hierarchically structured datasets using PyTorch“ (l.110), what is the benefit of using PyTorch here for simulation?
>
> We used PyTorch out of personal preference but it does not really matter which package is used as long as it allows seeded sampling. We will remove this detail from the methods.
>
> > Figure 1 needs to be improved in terms of overall readability: missing labels (panel C), unreadable fonts, neural networks are badly visible, and too small text in subplots. Make clear that D is only inference time, not training time. [...] Font sizes in all main figures are too small. In general, the resolution of the figures should be improved.
>
> Thank you for the suggestions, we will incorporate all of them in the final draft before the end of the rebuttals.
>
> > The importance sampling refinement is a nice idea. Showing results without this step to isolate its contribution would further improve the paper.
>
> Thanks! We will add ablation analyses to the SI after the reruns until the end of the rebuttals.
>
> > The code is not provided, preventing verification of the claim that open-source software and pretrained models are or will become available.
>
> The code *is* included in the originally submitted supplementary information zip file on openreview, but you are right that we should have made that more explicit. Thank you for the pointer.
>
> > The authors claim that prior work “at best nullifies the runtime advantage of NPE” (l. 68). How does this compare to the authors’ method?
>
> We have changed the related work section to soften this claim:
>
> *[in previous work] **the prior distribution is fixed at training time, requiring retraining whenever a user wishes to change the prior. This off-loads the amortization process to potential end-users, which strongly diminishes the runtime advantage of NPE for practical purposes.***
>
> Note that Arruda and Haberman do not provide standalone packages with pretrained models but note that their analyses are based on BayesFlow, i.e. the offered solutions are not directly actionable to the average end-user. This should not at all diminish the conceptual value of their contribution.
>
> > How can missing data be handled?
>
> This is indeed a prominent issue in statistical analysis and often boils down to the trade-off between two evils: (1) Missing data can sometimes be imputed from the remaining data but it therefore cannot add any information, and (2) some datasets are naturally spotty and if every row that has at least one element missing has to be discarded, potentially valuable information is lost and the inference may be biased towards the subpopulation with complete observations.
>
> For dataset preprocessing we have implemented a simple procedure that first checks for offending columns (those that have substantially more missing values than others) and removes them before discarding rows with NAs. On the other hand, masking missing values is a promising future extension to metabeta as it lends itself well to the transformer-based summary networks. On a larger level it is already implemented for the local summaries, as metabeta allows varying numbers of groups/subjects.
>
> > The paper claims full amortization, yet separate models are trained for different parameter dimensionalities. This weakens the claim of applicability for practitioners if the needed parameter dimensionality is not available. The authors should clarify early in the manuscript that amortization holds only for fixed dimensionality d and group structure q and that multiple models are trained
>
> Thanks for the remark! We have changed the following limitations section to address this.
>
> *Each trained version of metabeta is currently tailored to the size of the regression problem in
> terms of the number of fixed effects (d) and the number of random effects (q). This means there are
> potentially as many model versions as combinations of d and q. A single model with an upper bound
> to d and q is possible with our setup, but is likely to be outperformed by a model that is specialized for
> a given problem size. **The github repository provides pretrained versions of metabeta for several relevant parameter combinations. Together this collection of models acts like a single pretrained model, as each size can be pulled quickly from the repo for immediate deployment (as the model parameters weigh only around 20MB). From the practitioners perspective it makes no difference if there is a single or multiple pretrained models.***

---

> ### Author Response · Authors · 2025-11-22
>
> > How are priors incorporated as inputs: through direct parameterization or learned embeddings? What about priors which are not part of the training data?
>
> We add the following to the limitations and outlook section:
>
> *Currently, the prior families are fixed to HalfNormal for variance parameters and Normal for the rest. The parameters of the priors are concatenated to the summary vector s before being passed to the MLPs inside the normalizing flow. This approach could be generalized to varying prior families, whose identity can be embedded and simply added to the summary vector s. We plan to eventually extend metabeta to even more flexible prior specification.*
>
> > [...] relation to Whittle et al. “Distribution Transformers: Fast Approximate Bayesian Inference with On-The-Fly Prior Adaptation“
>
> Thanks for the suggestion, we add the following to the related works section:
>
> *Transformer-based architectures have emerged as a distinct line of research for amortized Bayesian inference. For instance Distribution Transformers (Whittle et al., 2025) represent prior and posterior as Gaussian Mixture Models whose parameters are mapped by transformers. A thorough comparison of transformer-based NPE methods was recently conducted by Mittal et al. (2025). These works demonstrate that transformer-based NPE can adapt efficiently to varying priors and heterogeneous datasets. However, they have not been tailored specifically to mixed-effects regression, and explicit incorporation of priors in NPE remains an active field of research.*

---

> > ### Comment · Reviewer_4rdr · 2025-11-25
> >
> > Thank you for the clarification regarding the prior. While you acknowledge the limitation in your outlook section, I think the paper would greatly benefit here from a more detailed analysis also of different priors, as the prior amortization is one of the main novelty claims in the paper.

---

> ### Comment · Reviewer_4rdr · 2025-11-25
>
> Thank you for improving the baseline method and addressing my major concern regarding the comparison with HMC.

---

### Official Review · Reviewer_dsuQ · 2025-10-28

**Soundness:** 2
**Presentation:** 2
**Contribution:** 2
**Rating:** 2
**Confidence:** 3

**Summary:**

This paper introduces a meta-learning framework for estimating the posterior distribution of coefficients in a Bayesian linear mixed-effects model, leveraging transformer architectures in a manner similar to TabPFN. To enhance the calibration of the predictive distribution, an additional post-hoc refinement stage—such as importance sampling or conformal prediction—is incorporated. The proposed approach is evaluated on both synthetic and semi-synthetic toy datasets.

**Strengths:**

Employing transformer-based meta-learning for diverse forms of amortized Bayesian inference is a compelling and important research direction. This work represents a valuable contribution to this growing area.

**Weaknesses:**

As noted by the authors, a key limitation is that a pretrained metabeta model can only be applied to datasets with a specific number of fixed effects and random effects. This constraint reduces the practical impact of the approach, as the model must be retrained for each new dataset with differing dimensionality. In such cases, the benefit of amortization becomes less compelling than using e.g., MCMC directly.

Another limitation is that all experiments are conducted on small-scale synthetic and semi-synthetic datasets, leaving the performance on real-world datasets uncertain.

**Questions:**

1. Could the authors describe the model details of the normalizing flow, particularly clarifying how the input s is utilized within the flow?

2. Did the authors employ both post-hoc refinement methods in the experiments, or was only one method used?

3. Could the authors provide experimental results on real-world datasets? Although the ground-truth effects are unknown, the model’s performance could be assessed by comparing the prediction accuracy of a linear mixed-effects model whose coefficients are estimated using metabeta or HMC.

---

> ### Author Response · Authors · 2025-11-21
>
> We thank the reviewer for engaging with our paper and deeming it a valuable contribution.
>
> > As noted by the authors, a key limitation is that a pretrained metabeta model can only be applied to datasets with a specific number of fixed effects and random effects. This constraint reduces the practical impact of the approach, as the model must be retrained for each new dataset with differing dimensionality. In such cases, the benefit of amortization becomes less compelling than using e.g., MCMC directly.
>
> We agree that the burden of training the model is not intended to be on the practitioner’s side and metabeta was never intended for that. We stress this in this update to the limitations section:
>
> *Each trained version of metabeta is currently tailored to the size of the regression problem in
> terms of the number of fixed effects (d) and the number of random effects (q). This means there are
> potentially as many model versions as combinations of d and q. A single model with an upper bound
> to d and q is possible with our setup, but is likely to be outperformed by a model that is specialized for
> a given problem size. The github repository provides pretrained versions of metabeta for several relevant parameter combinations. **Together this collection of models acts like a single pretrained model, as each size can be pulled quickly from the repo for immediate deployment (as the model parameters weigh only around 20MB). From the practitioners perspective it makes no difference if there is a single or multiple pretrained models.***
>
> > Another limitation is that all experiments are conducted on small-scale synthetic and semi-synthetic datasets, leaving the performance on real-world datasets uncertain. [...] Could the authors provide experimental results on real-world datasets?
>
> 1. Comparing model performance between metabeta and HMC is challenging when the true parameters are unknown. We will incorporate experiments that compare posterior predictive accuracy of both by the end of the rebuttals.
> 2. We are currently working on extending the training and test set using subsampling from real world datasets from [PMLB](https://github.com/EpistasisLab/pmlb) and [Simple Regression Models](https://proceedings.mlr.press/v58/lichtenberg17a.html). We will extend the current experiments with them by the end of the rebuttals.
>
> > Could the authors describe the model details of the normalizing flow, particularly clarifying how the input s is utilized within the flow?
>
> We use an affine coupling flow ([Dinh et al., 2017](https://arxiv.org/abs/1605.08803); [Papamakarios et al., 2021](https://arxiv.org/abs/1912.02762)). We first explain a single coupling step part of a normalizing flow block, and then explain the structure of each block. Let θ be the parameter vector and s be the summary vector produced by the corresponding summary network.
> 1. Partition θ in two parts.
> 2. Concatenate θ_1 with s and pass it through an MLP with residual connections.
> 3. The MLP maps this input on 2*k parameters of a pointwise affine function where k = length(θ_2).
> 4. Apply the affine function on θ_2 and concatenate the result with θ_1 to get the next “state” of θ.
> 5. Save the log det Jacobian of the pointwise affine function for the change of variables formula used for getting the exact density of the final transformed θ.
>
> Each coupling block starts with an Activation Normalization step, which applies an input-independent learned pointwise affine transform to θ. This is followed by a (fixed) random permutation of θ. Then the coupling step described above is done twice, such that both halves of θ are transformed. A normalizing flow consists of a sequence of coupling blocks, typically 3 to 6. Its expressiveness is a function of the size of the MLPs and of the number of coupling blocks. To make the summary vector s as informative as possible, we extend it with additional information like the priors and the dataset size.
>
> > Did the authors employ both post-hoc refinement methods in the experiments, or was only one method used?
>
> In all reported cases, both methods were used jointly as they are mutually inclusive.

---

> > ### Comment · Reviewer_dsuQ · 2025-11-25
> > **Thank you for your response**
> >
> > Thank you for your response which addressed some of my concerns! However, I am still concerned about its performance on real-world datasets beyond the simplified tabular synthetic data.

---

### Official Review · Reviewer_58pV · 2025-10-30

**Soundness:** 2
**Presentation:** 3
**Contribution:** 2
**Rating:** 2
**Confidence:** 4

**Summary:**

This paper proposes a method for neural amortized Bayesian inference (ABI) for linear mixed-effects (multilevel/hierarchical) models. The results are overall good but the contribution may be a bit small in light of existing literature. I am short on time due to the semester start. Apologies if my reviews are a bit short. I am happy to engage in reviewer discussion should be concerns not be clear.

**Strengths:**

- The paper works on an important topic.
- The applied and combined methods are sensible.
- The presentation is easy to follow for somone familiar with mixed-effects models.

**Weaknesses:**

- The contribution is overall small. Already previous papers provide ABI for multilevel models (and are correctly cited in the paper). The main addition here is the amortization over prior hyperparameters, but this has also been suggested in other places (https://arxiv.org/abs/2310.11122), althought admittedly not in a multilevel context.
- The HMC baseline seems to be incorrectly or at least not well implemented. For such simple multilevel models, HMC in PyMC or other PPLs should not struggle with any convergence or recovery issues. I assume your parameterization of the model wasn't quite right or optimal. Consider using a non-centered parameterization for the random effects. Or compare with an existing implementation of such model, e.g., via the brms R package using Stan as PPL backend, or bambi in python using PyMC as backend.
- The authors only focus on *linear* multilevel models where the error distribution is Gaussian. This unecessarily restricts the flexibility of the framework.
- No correlations between random effects are considered.
- The general formulation in terms of design matrices X and Z suggest the possibility of multiple grouping factors and corresponding random effects. Yet, your implementation just supports a single grouping factor as far as I can tell. I don't expect you to generalize your framework to multiple grouping factors right away. Just make this point more explicit.
- The term "transformer-based" perhaps oversells the point a bit that you use set transformer as summary networks.

**Questions:**

- The authors mention the aim of releasing a pretrained version of the model for free use. However, I am not sure if an "aim" is already a contribution of the paper. How far is the pretrained version?

---

> ### Author Response · Authors · 2025-11-21
>
> We thank the reviewer for engaging with our work, deeming our methods sensible and complimenting us on the presentation.
>
> > The contribution is overall small. Already previous papers provide ABI for multilevel models (and are correctly cited in the paper). The main addition here is the amortization over prior hyperparameters, but this has also been suggested in other places (https://arxiv.org/abs/2310.11122), although admittedly not in a multilevel context.
>
> We agree that the conceptual contribution is not groundbreaking. Yet, it comes with many engineering challenges that have to be solved, and our paper provides one demonstration for how that can be done (which is a valuable contribution in itself). Moreover, having a fast pipeline for Bayesian mixed-effects models that supports a wide range of generative models opens up new use cases and significantly lowers the barrier to adoption in applied settings. This is where we see the main value of our contribution.
>
> > I assume your parameterization of the model wasn't quite right or optimal. Consider using a non-centered parameterization for the random effects.
>
> The criticism is warranted and we have greatly improved the HMC fits by using NUTS instead of vanilla HMC, including non-centered parameterization of the random effects, and taking samples from four chains. Additionally, we save fit statistics like divergence counts, effective sample size and R-hat statistics and will add them to the SI.
>
> We have implemented three different setups: (1) one that matches the structure of the data generating process (distinguishing between fixed and random effects), (2) one with hierarchical parameterization that extends the official example by PyMC (distinguishing between mixed effects and pure fixed effects without additional random offsets), and (3) a Bambi-based setup with either default priors (specified by Bambi), true priors, and a hybrid version where the fixed effects priors are specified by Bambi and the others match the true priors. Hybrid versions of the two PyMC-based setups are included too, automatically specifying the priors for the fixed effects based on the moments of X and y (exactly matched with Bambi’s procedure). The table below compares each variant on a toy test set with 256 datasets in terms of parameter recovery (r, RMSE), average posterior width (SD) and average R-hat. The best values are marked bold.
>
> | Method | Parameter | r | RMSE | SD | R-hat |
> |-|-|-|-|-|-|
> | **Bambi: default** | fixed effects | **1.000** | 0.072 | 0.258 | 1.007 |
> | | random effects | **0.986** | 0.309 | 0.279 | 1.006 |
> | | variances | 0.994 | 0.141 | 0.219 | 1.006 |
> | **Bambi: true** | fixed effects | 0.999 | 0.110 | **0.231** | **1.003** |
> | | random effects | 0.984 | 0.327 | 0.258 | **1.004** |
> | | variances | **0.997** | 0.078 | 0.179 | 1.005 |
> | **Bambi: hybrid** | fixed effects | **1.000** | **0.062** | 0.237 | 1.004 |
> | | random effects | **0.986** | **0.307** | 0.263 | 1.005 |
> | | variances | **0.997** | **0.077** | **0.177** | 1.007 |
> | **PyMC: matched-true** | fixed effects | 0.999 | 0.113 | **0.230** | 1.004 |
> | | random effects | 0.984 | 0.327 | **0.257** | **1.004** |
> | | variances | 0.996 | 0.079 | 0.178 | **1.004** |
> | **PyMC: matched-hybrid** | fixed effects | **1.000** | **0.062** | 0.239 | 1.004 |
> | | random effects | **0.986** | **0.307** | 0.265 | 1.005 |
> | | variances | **0.997** | 0.078 | **0.177** | 1.006 |
> | **PyMC: hierarchical-true** | fixed effects | 0.999 | 0.119 | 0.231 | 1.005
> ||random effects | 0.984 | 0.330 | 0.258 | 1.005
> ||variances | 0.996 | 0.085 | 0.180 | **1.004**
> | **PyMC: hierarchical-hybrid** | fixed effects | **1.000** | 0.064 | 0.239 | 1.004 |
> | | random effects | **0.986** | **0.307** | 0.265 | 1.005 |
> | | variances | 0.996 | 0.079 | 0.178 | **1.004** |
>
> This marks the hybrid versions of Bambi and PyMC (matched) as the top contenders. The next table compares the divergence behavior (mean number of divergent samples, frequency of any divergent sample occurring) over variants:
>
> | Method | mean | frequency |
> |-|-|-|
> | **Bambi: default** | 2.394 | 0.254
> | **Bambi: true** | **0.971** | 0.193
> | **Bambi: hybrid** | 1.473 | 0.187
> | **PyMC: matched-true** | 1.576 | 0.178
> | **PyMC: matched-hybrid** | 2.185 | **0.165**
> | **PyMC: hierarchical-true** | 1.462 | 0.175
> | **PyMC: hierarchical-hybrid** | 1.699 | 0.186
>
> This suggests, the PyMC-hybrid configuration has the rarest divergence occurrence (but when it diverges, it has a relatively high number of divergent samples, with 2 outlier datasets out of 256). On the other hand, hybrid Bambi has a relatively higher divergence probability (but when it diverges, it produces relatively fewer divergent samples). We believe this overall favors PyMC-matched-hybrid. For fair comparisons, datasets with high divergence will be removed from the test set. We will rerun all experiments with the improved implementation of HMC and will also report diagnostics in the SI until the end of the rebuttal phase.

---

> > ### Author Response · Authors · 2025-11-21
> >
> > > The authors only focus on linear multilevel models where the error distribution is Gaussian. This unnecessarily restricts the flexibility of the framework. No correlations between random effects are considered.
> >
> > Thanks for this comment. We agree that this is a limitation, but also an exciting avenue for future work, which we now discuss in our limitations and outlook section:
> >
> > *Our framework is currently specialized on Bayesian linear mixed effects regression, but the required steps to generalized mixed-effects models are in parts small: Data simulation would require an additional response function around the linear term. The response function type could be passed to metabeta along with the priors, letting the network condition its computation on it. Extending importance sampling for non-linear cases is non-trivial. Overall, this generalization is an exciting direction for future work.*
> >
> > That being said, linear mixed-effects models remain the most widely used class of mixed-effects models. Thus, if one aims for practical relevance, they provide a natural starting point. Similarly, using uncorrelated random effects is a proof of concept starting point and extending the framework to correlated random effects is both relatively straight-forward and a midterm goal.
> >
> > > Your implementation just supports a single grouping factor as far as I can tell. I don't expect you to generalize your framework to multiple grouping factors right away. Just make this point more explicit.
> >
> > Good catch, you are right that the hierarchical NPE architecture only supports a single parallel grouping factor. We have added the following to the limitation section:
> >
> > *Hierarchical NPE is well suited for hierarchical regression with one grouping factor: Multiple parallel grouping factors would require non-trivial extensions to dataset summarization and integration of multiple summaries. However, it is conceptually straightforward to extend the framework to multiple nested grouping factors (e.g. schools and classrooms within schools), and remains an exciting avenue for future developments.*
> >
> > > The term "transformer-based" perhaps oversells the point a bit that you use set transformer as summary networks.
> >
> > Thank you for this comment. We have decided to replace the *transformed-based* label with the more generic *neural network-based*.
> >
> > > The authors mention the aim of releasing a pretrained version of the model for free use. However, I am not sure if an "aim" is already a contribution of the paper. How far is the pretrained version?
> >
> > Thank you for raising this point. To clarify: the pretrained models are not just an aim but will be part of the submission once all links are uncensored. We include them in the github repository, thus, the pre-trained models are ready for immediate use.

---

> > > ### Comment · Reviewer_58pV · 2025-11-26
> > >
> > > Thank you for your responses. I appreciate your efforts and raised my score to 4.

---

### Official Review · Reviewer_xGrJ · 2025-10-30

**Soundness:** 2
**Presentation:** 2
**Contribution:** 3
**Rating:** 2
**Confidence:** 4

**Summary:**

The paper proposes metabeta, an amortized Bayesian inference framework for mixed-effects
(hierarchical) regression. The method draws on techniques from simulation-based
inference (SBI), particularly neural posterior estimation (NPE) with normalizing flows
and permutation-invariant set encoders. By pretraining on many simulated hierarchical
datasets under varying priors, metabeta learns to approximate posteriors over both
global and local parameters of linear mixed-effects models (LMEs). At inference time,
users can provide their own priors and data, and the network outputs approximate
posteriors within seconds—potentially replacing MCMC for common hierarchical modeling
tasks. To improve calibration, the authors add post-hoc importance sampling (IS) using
the analytic LME likelihood and conformal prediction to adjust coverage. Experiments
compare metabeta to Hamiltonian Monte Carlo (HMC) on toy and real data, showing large
speed gains and competitive accuracy.

**Strengths:**

- Timely and practically motivated: The work targets a real bottleneck (computational
  cost of Bayesian mixed-effects regression) and adapts amortized SBI tools for this
  setting. This is an interesting application domain for amortized inference.
- Clear architecture design: The hierarchical set-transformer encoder combined with
  flow-based posterior heads is a reasonable and interpretable choice. The post-hoc
  importance sampling and conformal calibration steps are conceptually clean and
  computationally lightweight.
- Empirical performance: On toy and real hierarchical datasets, metabeta achieves
  parameter recovery and coverage comparable to HMC at a fraction of inference time. The
  results demonstrate that amortized models can produce usable posterior approximations
  in classical regression problems.
- Potential impact: If validated under fair comparison, such amortized inference could
  make Bayesian mixed-effects modeling far more accessible to applied fields (social
  sciences, bioinformatics, etc.) where MCMC remains the default.

**Weaknesses:**

### Conceptual framing and related work

- The paper is not truly an SBI setting: the LME simulator and likelihood are
  analytically known. metabeta uses SBI tools for amortized efficiency, not because
  inference is likelihood-free. This distinction should be made explicit and more prominent.
- The historical narrative around NPE and BayesFlow is inaccurate. Neural Posterior
  Estimation (also the amortized version) was introduced by Papamakarios et al. (2016) and extended by Lueckmann et
  al. (2017) and Greenberg et al. (2019). BayesFlow (Radev et al., 2020) later provided
  a practical amortized inference framework with set encoders, not transformers.
  Transformer-based amortized inference (e.g., Whittle et al., 2025; Mittal et al.,
  2025; Reuter et al., 2025) represents a distinct and more recent line of work. The
  related work section should reflect this chronology.
- The related work section omits key hierarchical SBI approaches such as Rodrigues et
  al. (NeurIPS 2021). The absence of these citations distorts context and weakens the
  claim of novelty.

### Methodology and baselines

- The HMC comparison is potentially unfair. Divergences in HMC typically signal poor
  tuning rather than algorithmic failure. [Non-centered parameterizations](https://sjster.github.io/introduction_to_computational_statistics/docs/Production/Reparameterization.html), robust
  step-size tuning, multiple chains, and R-hat diagnostics are standard. The authors
  should verify that best practices were followed; otherwise, the claimed accuracy
  advantage is not meaningful.
- Missing other baselines for hierarchical inference: The comparison currently focuses on HMC, but omits several established fast Bayesian or approximate inference methods that directly apply to mixed-effects models. For example: Variational Inference (VI) offers scalable approximate posteriors and would provide a relevant amortized or single-dataset baseline. INLA (Integrated Nested Laplace Approximation), a deterministic and highly efficient method for latent Gaussian models, widely used for Bayesian mixed-effects and spatial models; often matches MCMC accuracy at a fraction of the cost. Laplace / GLMM approximations — the classical second-order Gaussian approximation around the MAP, as implemented in standard GLMM software. Including these would contextualize metabeta's speed and accuracy gains relative to well-known fast alternatives rather than only to a potentially under-tuned HMC baseline.
- The posterior quality is inconsistent: in the toy example (Fig. 1C), metabeta produces irregular shapes for otherwise Gaussian-like posteriors. This raises questions about the flow architecture and whether the networks are overflexible or underregularized.
- Importance sampling is introduced without sufficient justification. If NPE is trained
  on the correct generative model, IS should not be required. The authors should explain
  whether IS corrects training–prior mismatch or residual amortization bias. Similar
  post-hoc IS refinements have been proposed in SBI (e.g., Dax et al., 2023) and should
  be cited.
- Evaluation metrics focus on parameter RMSE and correlation, which are not appropriate
  for Bayesian inference. The true parameter need not coincide with posterior means.
  Calibration metrics such as simulation-based calibration (SBC) or log-probability of
  the true parameters would be more informative.
- Runtime comparison is anecdotal (“orders of magnitude faster”) and lacks hardware
  disclosure or amortized cost estimates. Practitioners need wall-clock times on
  standardized setups.

### Presentation and reproducibility

- Figure 1 is difficult to interpret; caption variables do not align with figure
  notation. Posterior density plots based on kernel estimates obscure the fact that the
  NPE model defines a parametric PDF.
- Quantitative results: Table 1 lacks error bars or multiple-seed repetitions, making it
  unclear how stable the reported metrics are.
- Coverage statements (“good coverage”) are qualitative; comparative or numerical values
  are needed.
- The software contribution is underspecified. Code is “hidden” for review, but an
  anonymous repository is easily possible. Since the method’s accessibility depends on
  this, it should be part of the submission.
- Tone in the introduction (“prohibitively long inference times”) overstates the
  limitations of MCMC.

**Questions:**

1) Clarification of training data generation: How are the simulated training datasets
   (X, Z) constructed? Are they drawn from distributions intended to mimic real-world
   predictors, or are they generic Gaussian designs? How does the method generalize if
   real data differ strongly from the training simulations?
2) Role of real data: If the model is trained entirely on synthetic data, what exactly
   is the pipeline for applying it to real datasets? Are priors and covariate
   distributions assumed to match?
3) Calibration discrepancy: The paper attributes poor calibration to the forward-KL
   objective’s mass-covering tendency, whereas SBI literature (e.g., Hermans et al.
   2022), often finds flows too narrow. Can the authors reconcile this difference? Might
   the issue arise from too little training data or too simplistic flows?
4) Alternative posterior networks: Could score-based or flow-matching estimators
   mitigate the need for IS and conformal calibration? These avoid the forward-KL bias
   entirely.
5) Importance sampling diagnostics: What are the effective sample sizes and weight
   variances of the IS correction? Without them, it is hard to assess stability.
6) Scope beyond Gaussian LMEs: Does metabeta handle generalized mixed-effects models
   (e.g., logistic, Poisson)? If not, how would the method need to change to support
   non-Gaussian likelihoods?

---

> ### Author Response · Authors · 2025-11-21
>
> We thank the reviewer for their thoughtful and in-depth review of our paper. We are glad the reviewer found our potential contribution relevant and liked the design choices.
>
> > The paper is not truly an SBI setting: the LME simulator and likelihood are analytically known. metabeta uses SBI tools for amortized efficiency, not because inference is likelihood-free. This distinction should be made explicit and more prominent.
>
> Thank you for this comment, which we fully agree with. To make this more explicit, we have added the following to the related works section:
>
> *In contrast to simulation-based inference, our problem setting is not likelihood-free. We employ neural posterior estimation for amortized efficiency and to enable fast, reusable Bayesian inference across datasets, rather than to approximate an intractable likelihood.*
>
> > The historical narrative around NPE and BayesFlow is inaccurate [...]
>
> Thanks for the useful pointers, we have incorporated them into the related works and model architecture section as follows:
>
> ***Neural posterior estimation (NPE) — the simulation-based amortization of a neural network posterior — has a long and well-established history. Early work by Papamakarios & Murray (2016) introduced neural conditional density estimators for directly approximating posteriors from simulations. This approach was extended by Lueckmann et al. (2017), who incorporated importance weighting to enable sequential refinement of posterior approximations, and by Greenberg et al. (2019), who proposed automatic posterior transformation (APT), increasing flexibility in proposal adaptation and posterior modeling. These methods form the core foundations of amortized simulation-based Bayesian inference.***
>
> *Rezende & Mohamed (2015) pioneered the use of conditional normalizing flows (Papamakarios et al., 2021; Kobyzev et al., 2021) for amortized inference. Together with Gordon et al. (2018), they laid the groundwork for BayesFlow (Radev et al., 2020, 2023),* ***which introduced a practical workflow for globally amortized Bayesian inference using summary encoders and normalizing flows. Subsequent extensions adapted BayesFlow to hierarchical Bayesian models*** *(Habermann et al., 2024) and to non-linear mixed-effects models for cell biology and pharmacology (Arruda et al., 2023). In both cases,* ***the prior distribution is fixed at training time, requiring retraining whenever a user wishes to change the prior. This off-loads the amortization process to potential end-users, which strongly diminishes the runtime advantage of NPE for practical purposes.***
>
> ***More recently, transformer-based architectures have emerged as a distinct line of research for amortized Bayesian inference. For instance Distribution Transformers (Whittle et al., 2025) represent prior and posterior as Gaussian Mixture Models whose parameters are mapped by transformers. A thorough comparison of transformer-based NPE methods was recently conducted by Mittal et al. (2025). These works demonstrate that transformer-based NPE can adapt efficiently to varying priors and heterogeneous datasets. However, they have not been tailored specifically to mixed-effects regression, and explicit incorporation of priors in NPE remains an active field of research.***
>
> [...]
>
> *Inference on global and local parameters is separated for* ***hierarchical NPE (Rodrigues et al 2021; Heinrich et al., 2023).***

---

> > ### Author Response · Authors · 2025-11-21
> >
> > > Non-centered parameterizations, robust step-size tuning, multiple chains, and R-hat diagnostics are standard. The authors should verify that best practices were followed:
> >
> > The criticism is warranted and we have greatly improved the HMC fits by using NUTS instead of vanilla HMC, including non-centered parameterization of the random effects, and taking samples from four chains. Additionally, we save fit statistics like divergence counts, effective sample size and R-hat statistics and will add them to the SI.
> >
> > We have implemented three different setups: (1) one that matches the structure of the data generating process (distinguishing between fixed and random effects), (2) one with hierarchical parameterization that extends the official example by PyMC (distinguishing between mixed effects and pure fixed effects without additional random offsets), and (3) a Bambi-based setup with either default priors (specified by Bambi), true priors, and a hybrid version where the fixed effects priors are specified by Bambi and the others match the true priors. Hybrid versions of the two PyMC-based setups are included too, automatically specifying the priors for the fixed effects based on the moments of X and y (exactly matched with Bambi’s procedure). The table below compares each variant on a toy test set with 256 datasets in terms of parameter recovery (r, RMSE), average posterior width (SD) and average R-hat. The best values are marked bold.
> >
> > | Method | Parameter | r | RMSE | SD | R-hat |
> > |-|-|-|-|-|-|
> > | **Bambi: default** | fixed effects | **1.000** | 0.072 | 0.258 | 1.007 |
> > | | random effects | **0.986** | 0.309 | 0.279 | 1.006 |
> > | | variances | 0.994 | 0.141 | 0.219 | 1.006 |
> > | **Bambi: true** | fixed effects | 0.999 | 0.110 | **0.231** | **1.003** |
> > | | random effects | 0.984 | 0.327 | 0.258 | **1.004** |
> > | | variances | **0.997** | 0.078 | 0.179 | 1.005 |
> > | **Bambi: hybrid** | fixed effects | **1.000** | **0.062** | 0.237 | 1.004 |
> > | | random effects | **0.986** | **0.307** | 0.263 | 1.005 |
> > | | variances | **0.997** | **0.077** | **0.177** | 1.007 |
> > | **PyMC: matched-true** | fixed effects | 0.999 | 0.113 | **0.230** | 1.004 |
> > | | random effects | 0.984 | 0.327 | **0.257** | **1.004** |
> > | | variances | 0.996 | 0.079 | 0.178 | **1.004** |
> > | **PyMC: matched-hybrid** | fixed effects | **1.000** | **0.062** | 0.239 | 1.004 |
> > | | random effects | **0.986** | **0.307** | 0.265 | 1.005 |
> > | | variances | **0.997** | 0.078 | **0.177** | 1.006 |
> > | **PyMC: hierarchical-true** | fixed effects | 0.999 | 0.119 | 0.231 | 1.005
> > ||random effects | 0.984 | 0.330 | 0.258 | 1.005
> > ||variances | 0.996 | 0.085 | 0.180 | **1.004**
> > | **PyMC: hierarchical-hybrid** | fixed effects | **1.000** | 0.064 | 0.239 | 1.004 |
> > | | random effects | **0.986** | **0.307** | 0.265 | 1.005 |
> > | | variances | 0.996 | 0.079 | 0.178 | **1.004** |
> >
> > This marks the hybrid versions of Bambi and PyMC (matched) as the top contenders. The next table compares the divergence behavior (mean number of divergent samples, frequency of any divergent sample occurring) over variants:
> >
> > | Method | mean | frequency |
> > |-|-|-|
> > | **Bambi: default** | 2.394 | 0.254
> > | **Bambi: true** | **0.971** | 0.193
> > | **Bambi: hybrid** | 1.473 | 0.187
> > | **PyMC: matched-true** | 1.576 | 0.178
> > | **PyMC: matched-hybrid** | 2.185 | **0.165**
> > | **PyMC: hierarchical-true** | 1.462 | 0.175
> > | **PyMC: hierarchical-hybrid** | 1.699 | 0.186
> >
> > This suggests, the PyMC-hybrid configuration has the rarest divergence occurrence (but when it diverges, it has a relatively high number of divergent samples, with 2 outlier datasets out of 256). On the other hand, hybrid Bambi has a relatively higher divergence probability (but when it diverges, it produces relatively fewer divergent samples). We believe this overall favors PyMC-matched-hybrid. For fair comparisons, datasets with high divergence will be removed from the test set. **We will rerun all experiments with the improved implementation of HMC and will also report diagnostics in the SI until the end of the rebuttal phase.**

---

> ### Author Response · Authors · 2025-11-21
>
> > Variational Inference (VI) offers scalable approximate posteriors and would provide a relevant amortized or single-dataset baseline [...]
>
> We agree and have extended the comparisons to ADVI via PyMC or Bambi. We use the best performing Bambi and PyMC setup from above and compare three configurations for ADVI: (1) the default one, (2) a short one with 25k iterations and Adam with lr=1e-2, (3) a longer one with 100k iterations and Adam with lr=1e-3.
>
> | Method | Parameter | r | RMSE | SD |
> |-|-|-|-|-|
> | **Bambi: default** | fixed effects | 0.908 | 1.997 | 0.308 | 0.000
> ||random effects | 0.610 | 2.041 | 0.482 | 0.000
> ||variances | 0.745 | 1.214 | 0.449 | 0.000
> | **Bambi: short** | fixed effects | 0.999 | 0.128 | 0.065 | 0.000
> || random effects | 0.984 | 0.334 | **0.143** | 0.000
> || variances | 0.996 | 0.174 | **0.054** | 0.000
> | **Bambi: long** | fixed effects | 0.999 | 0.121 | 0.065 | 0.000
> || random effects | **0.985** | **0.320** | 0.145 | 0.000
> || variances | **0.997** | **0.100** | 0.055 | 0.000
> | **PyMC: default** | fixed effects | 0.909 | 1.997 | 0.307 | 0.000
> ||random effects | 0.610 | 2.041 | 0.481 | 0.000
> ||variances | 0.746 | 1.210 | 0.448 | 0.000
> | **PyMC: short** | fixed effects | **1.000** | 0.099 | **0.064** | 0.000
> || random effects | 0.983 | 0.340 | 0.144 | 0.000
> || variances | 0.995 | 0.176 | 0.053 | 0.000
> | **PyMC: long** | fixed effects | **1.000** | **0.096** | 0.065 | 0.000
> || random effects | **0.985** | 0.321 | 0.145 | 0.000
> || variances | **0.997** | **0.100** | 0.055 | 0.000
>
> The PyMC-long configuration has the overall edge among the tested VI setups. It performs slightly worse than its MCMC counterpart but only by a small margin, making it suitable as an additional competitor model. We will include it for the large-scale reruns.
>
>
> > The posterior quality is inconsistent: in the toy example (Fig. 1C), metabeta produces irregular shapes for otherwise Gaussian-like posteriors. This raises questions about the flow architecture and whether the networks are overflexible or underregularized.
>
> The statement is correct in that the two posteriors do not exactly match - however the Gaussian-like posteriors are produced by metabeta as indicated by the legend in the figure. The irregularity in the HMC posterior was produced by the suboptimal specification of the MCMC model which we have fixed in our updated version and will update Figure 1 during next week.
>
> > If NPE is trained on the correct generative model, IS should not be required. [...] The authors should explain whether IS corrects training–prior mismatch or residual amortization bias. Similar post-hoc IS refinements have been proposed in SBI (e.g., Dax et al., 2023) and should be cited.
>
> We agree with the reviewer that it is important to give a brief justification for the importance sampling post-hoc refinement. We have therefore added the following to our manuscript:
>
> ***In the idealized limit of infinite network capacity, neural posterior flexibility, infinite simulations, and perfectly converged optimization, metabeta would not require any further correction. However, in practice these conditions are never fully met.*** [...] *Thus, we can use importance sampling to improve posterior estimation (Tokdar & Kass, 2010; Dax et al., 2023)* ***to correct for inaccuracies of the amortized estimator.***
>
> Dax et al., 2023 is already cited in our original manuscript as seen above.
>
> > Evaluation metrics focus on parameter RMSE and correlation, which are not appropriate for Bayesian inference. [...] Calibration metrics such as simulation-based calibration (SBC) or log-probability of the true parameters would be more informative.
>
> Practitioners base decisions on point estimates (and should ideally also incorporate uncertainty estimates), which justifies correlation and RMSE as general evaluation metrics for parameter recovery. We would love to compare the densities of the true parameters under each posterior but it is not so trivial from a technical perspective: Unfortunately, **PyMC only offers to evaluate the unnormalized posterior density of the true parameters (e.g. with model.compile_logp()). Computing the additionally required marginal log p(y) is not straightforward and comes with its own caveats. We welcome your ideas and suggestions on how to solve this elegantly.**
>
> Simulation based calibration is usually evaluated visually (using SBC histograms or ECDFs) - while quantitative comparisons (e.g. with the Wasserstein-1-distance to the standard uniform distribution) are possible, their interpretation is less intuitive. We will add SBC plot comparisons to the SI until the end of the rebuttals.

---

> ### Author Response · Authors · 2025-11-21
>
> > Runtime comparison is anecdotal (“orders of magnitude faster”) and lacks hardware disclosure or amortized cost estimates. Practitioners need wall-clock times on standardized setups.
>
> Thank you for this helpful feedback. We agree that runtime comparisons should be transparent and reproducible. Reported wall-clock times were measured on a MacBook Air M2 with 24 GB of RAM (as noted in the Figure 1 caption), which we believe is a reasonably representative setup for many practitioners. In addition to the average runtimes shown in Figure 1D, we will include a table of per-dataset runtimes (for both metabeta and HMC) during the next week.
>
> > Figure 1 is difficult to interpret; caption variables do not align with figure notation.
>
> We have carefully examined the variables used in the figure and found no mismatch. Could you please clarify, where exactly there is misalignment? We are happy to incorporate your suggestions.
>
>  > Posterior density plots based on kernel estimates obscure the fact that the NPE model defines a parametric PDF.
>
> Correct, the NPE model allows evaluating the joint posterior density of any appropriately sized vector. However because it is the joint density, exhaustive search of the parameter space is prohibitively expensive, hence we sample from the posterior (which is computationally cheap) and calculate a KDE approximation of the marginal densities. While mathematically cleaner, direct evaluation of the marginal posterior densities is impractical and should produce very similar visuals. If you insist, we can re-do the posterior plots that way.
>
> > Quantitative results: Table 1 lacks error bars or multiple-seed repetitions, making it unclear how stable the reported metrics are.
>
> We have started re-running the experiments with additional seeds and will add the results until the end of the rebuttals.
>
> > Coverage statements (“good coverage”) are qualitative; comparative or numerical values are needed.
>
> Thank you for the suggestion. We already provide quantification of coverage using coverage error, CE(alpha), both visually and numerically (see Tables and Fig 2B). We will explicitly mention the corresponding average coverage errors whenever reporting qualitative values in the text, for example:
>
> *Our model again has [...] better coverage (Figure 2B, mean CE =  x1 vs. x2). This is notably reflected in its credible intervals (Figure 3A).*
>
> The x's will be replaced by actual values after the rerun is complete.
>
> > The software contribution is underspecified. Code is “hidden” for review .
>
> Thank you for raising this important point. We had already uploaded a .zip file with all code alongside our original submission on OpenReview. A GitHub repository already exists but was retracted for anonymity as stated on page 2. We will of course update this link in an eventual de-anonymized version.
>
> > Tone in the introduction (“prohibitively long inference times”) overstates the limitations of MCMC.
>
> Based on this suggestion, we have reworded this sentence to:
>
> *From a practitioner’s perspective, this is undesirable, as MCMC typically requires significant runtime, even for moderately sized datasets.*
>
> > How are the simulated training datasets (X, Z) constructed? [...] If the model is trained entirely on synthetic data, what exactly is the pipeline for applying it to real datasets?
>
> The simulation process is described in detail in Appendix A. In short, the columns of X are sampled from various distributions intended to be diverse, including random cutoffs, skewed data, as well as dependence between columns. If you wish, we can move these details to the main text.
>
> Of course, NPE performance heavily depends on the diversity of its training data (e.g. a model trained on purely uniform X will struggle to transfer to purely Gaussian X). Note that for the four semi-synthetic test sets, the columns are subsampled from real mixed-effects datasets often used in educational settings. In order to further increase distributional diversity, we are currently working on extending the training set by a similar sampling mechanism from real datasets from [Simple Regression Models](https://proceedings.mlr.press/v58/lichtenberg17a.html) and [PMLB](https://github.com/EpistasisLab/pmlb) and will incorporate the results until the end of the rebuttals.

---

> ### Author Response · Authors · 2025-11-21
>
> > The paper attributes poor calibration to the forward-KL objective’s mass-covering tendency, whereas SBI literature (e.g., Hermans et al. 2022), often finds flows too narrow.
>
> We have checked the suggested paper and its [repository](https://github.com/montefiore-institute/trust-crisis-in-simulation-based-inference?tab=readme-ov-file). The plot suggests that the coverage is indeed too narrow for several SBI challenges. However, the details of the used NPE algorithm are obscured - a look at the repository suggests that for SNPE the [sbi toolbox](https://sbi.readthedocs.io/en/latest/tutorials/16_implemented_methods.html#neural-posterior-estimation-npe) was used. The current implementation of NPE by this toolbox follows Greenberg et al. (2019 at ICML). It is unclear whether the implementation used to be affine coupling when the paper was written. Our own observation of too wide “vanilla” posteriors match what is reported in [Chen et al. (2025)](https://arxiv.org/pdf/2410.19105), who used affine coupling flows as well.
>
> > Could score-based or flow-matching estimators mitigate the need for IS and conformal calibration? These avoid the forward-KL bias entirely.
>
> Thank you for this suggestion. Other NPE methods are promising options with potentially better accuracy and less need for posthoc refinement. We already discuss the potential use of continuous normalizing flows (e.g. conditional diffusion or flow matching) instead of affine coupling in section 4.1 and have made the following changes:
>
> *Our choice of model architecture trades of posterior expressivity for computation speed: Other
> normalizing flow methods like [...] Flow Matching (Wildberger et al., 2023), Conditional Diffusions (Chen et al., 2025; Reuter et al., 2025) [...] offer more flexible posterior shapes, but posterior sampling is considerably more expensive than for Affine Coupling Flows: **Sampling from Conditional diffusion requires solving a stochastic differential equation, and sampling from Flow Matching requires numerical integration. Also, substantially more network evaluations are necessary, whereas for discrete Normalizing Flows, one evaluation suffices. This would strongly diminish the speed advantage in comparison to standard approaches like MCMC.***
>
> Continuous normalizing flows remain an interesting alternative and we hope to incorporate it into metabeta if sampling from them becomes cheaper.
>
> > What are the effective sample sizes and weight variances of the IS correction? Without them, it is hard to assess stability.
>
> We will include IS diagnostics and a parameter recovery comparison (IS vs. no IS) in the SI until the end of the rebuttals.
>
> > Does metabeta handle generalized mixed-effects models (e.g., logistic, Poisson)? If not, how would the method need to change to support non-Gaussian likelihoods?
>
> Thanks for this question. metabeta does not handle generalized mixed-effects models at present, but we agree that this is an exciting direction for future work, and have expanded our discussion section to highlight this:
>
> *Our framework is currently specialized on Bayesian linear mixed effects regression, but the required steps to generalized mixed-effects models are in parts small: Data simulation would require an additional response function around the linear term. The response function type could be passed to metabeta along with the priors, letting the network condition its computation on it. Extending importance sampling for non-linear cases is non-trivial. Overall, this generalization is an exciting direction for future work.*

---

> > ### Comment · Reviewer_xGrJ · 2025-11-26
> > **Response to rebuttal**
> >
> > I thank the authors for the detailed rebuttal. I appreciate the detailed comparisons of the MCMC baseline settings.
> >
> > Clarification for figure 1: my point was that the caption text does mention some of the crucial symbols used in the figure, making it harder to follow through the figure.
> >
> > Clarification on NPE evaluation: there might be a misunderstanding, but as I see it, it’s perfectly doing to evaluate large vectors of samples under the conditional flow used for NPE, e.g., it’s as fast as sampling. Once you have the samples and the corresponding log probs you can nicely visualise them using coloured  contour plots in 2D or density plots in 1D.
> >
> > I am looking forward to seeing the  additional results, especially the comparisons with more grounded MCMC and VI baselines vs metabeta.

---

### Author Response · Authors · 2025-12-04

We thank all reviewers for their constructive and helpful feedback. Their input was immensely valuable to further improve our submission. In response to the reviewers' feedback, we have made the following major additions:
- We vastly improved stability and performance of HMC fits and discard datasets with significant divergence or mixing problems.
- We added Variational Inference (ADVI) as a competitive model that is faster than HMC, as it is the most commonly used alternative for MCMC.
- We substantially extended data simulation, incorporating data from 271 datasets from two large benchmarks for tabular data (SRM and PMLB).
- We retrained our models and repeated all experiments with the new data, new model and new competitive fits.
- We evaluated all models on multiple out of distribution hierarchical regression sets without known parameters.
- We included SBC plots, posterior accuracy and principled run time evaluations for various regression problem sizes.

We have updated our submitted pdf accordingly. We again want to thank the reviewers for their valuable time, attention and for actively taking part in the review process.

---

### Meta-Review · Area_Chair_WrLg · 2026-01-08

**Summary:**

The reviewers made many suggestions for strengthening the experimental section of the paper. The authors engaged very well with this (see their post), and this is a crucial contribution of the paper, given that amortized approximate inference is a well-established idea by now.

The revisions go beyond clarification and discussion. It has been a grey area as to whether papers that have been significantly altered should be accepted, or reviewed again as a new conference submission. Given the reviewer's assessments (some did respond directly, others seem to not have been able to respond in time), another round of reviewing is appropriate here.

**Reviewer Concerns:**

It is clear that the authors have put much additional effort into taking on board the reviewers' concerns. It seems that the paper has become much stronger as a consequence.

**Reviewer Scores:**

Several reviewers engaged carefully with the authors, and perhaps would have raised their scores. However, this is in response to major additions to the paper. These revisions go beyond clarification and discussion. It has been a grey area as to whether papers that have been significantly altered should be accepted. Other reviewers have engaged, and listed their final recommendations.

---

### Decision · Program_Chairs · 2026-01-26

Reject